# Lookahead Sample Reward Guidance for Test-Time Scaling of Diffusion Models

**Yeongmin Kim** [1]  **Donghyeok Shin** [1]  **Byeonghu Na** [1]  **Minsang Park** [1]  **Richard Lee Kim** [1]  **Il-Chul Moon** [1 2]

## Abstract

Diffusion models have demonstrated strong generative performance; however, generated samples often fail to fully align with human intent. This paper studies an efficient test-time scaling method for sampling from regions with higher human-aligned reward values. Existing methods for computing the expected future reward (EFR) face important limitations: backward rollout incurs prohibitively high sampling costs, while Tweedie-based approaches, including Sequential Monte Carlo and gradient guidance, suffer from bias and inherent sampling issues. We show that the EFR at any $\mathbf{x}_t$ can be computed using only marginal samples from a pre-trained diffusion model, enabling closed-form reward guidance without neural backpropagation. To further improve efficiency, we introduce a few-step lookahead sampling and an accurate solver that guides particles toward high-reward lookahead samples. We refer to this sampling scheme as LiDAR sampling. LiDAR achieves the same GenEval performance as the latest gradient guidance method for SDXL with a $9.5\times$ speedup. We release the code at https://github.com/aailab-kaist/Diffusion-LiDAR-Sampling.

## 1. Introduction

Diffusion models (Sohl-Dickstein et al., 2015; Ho et al., 2020; Song et al., 2021b) have recently emerged as a powerful paradigm for sample generation, demonstrating strong performance across a wide range of domains, including images (Podell et al., 2024; Esser et al., 2024), videos (Ho et al., 2022), and language (Lou et al., 2024; Nie et al., 2025). Despite these advances, the generated samples often fail to

*Table 1.* Comparison of test-time scaling methods for diffusion models. *Efficient-Rollout* indicates whether EFR estimation avoids per-timestep rollout sampling; *Finite i.i.d.* indicates whether performance remains consistent under a finite number of target sampling particles; *No-Taylor* indicates whether reward feedback can be obtained without Taylor approximation; and *No-BackPropagation* indicates whether neural differentiation is required.

| Method | Backward Rollout | Sequential Monte Carlo | Gradient Guidance | LiDAR (Ours) |
|---|---|---|---|---|
| | (Holderrieth et al., 2026) | (Singhal et al., 2025) | (Bansal et al., 2024) | |
| | (Potaptchik et al., 2025) | (Li et al., 2025) | (Na et al., 2025) | |
| Effi.-Rollout | ✗ | ✓ | ✓ | ✓ |
| Finite i.i.d. | ✓ | ✗ | ✓ | ✓ |
| No-Taylor | ✓ | ✗ | ✗ | ✓ |
| No-BackPro. | △ | ✓ | ✗ | ✓ |

fully align with user intent. For instance, prompt–image misalignment remains a well-recognized challenge in text-to-image diffusion models (Na et al., 2025).

To address this issue, reward functions that capture human intent (Xu et al., 2023a; Ma et al., 2025) have been incorporated into generative modeling pipelines, either through fine-tuning (Black et al., 2024; Liu et al., 2025; Wu et al., 2024; Kim et al., 2025b) or via test-time scaling (Kim et al., 2023; Bansal et al., 2024; Na et al., 2025; Singhal et al., 2025; Li et al., 2025). Although these two approaches are orthogonal, test-time scaling methods have attracted particular attention because they can achieve substantial performance gains without additional training cost. However, these methods increase inference-time computation; thus, improving the performance gain per computation unit is a key factor in achieving better compute-optimal performance (Snell et al., 2025).

While rewards are typically assigned to the final generated sample $\mathbf{x}_0$, the target distribution is often defined as a reward-tilted distribution (i.e., $p_{\boldsymbol{\theta}}^r(\mathbf{x}_0) \propto p_{\boldsymbol{\theta}}(\mathbf{x}_0) \exp(r(\mathbf{x}_0))$). A major challenge in test-time scaling lies in computing the *Expected Future Reward* (EFR) for an intermediate particle $\mathbf{x}_t$ and incorporating it into the iterative sampling process. Although directly computing the EFR through backward rollout is computationally expensive, recent methods leverage Taylor approximations to compute the EFR in a relatively efficient manner (Chung et al., 2023; Bansal et al., 2024; Singhal et al., 2025). However,

[1]Korea Advanced Institute of Science and Technology (KAIST), Daejeon, Republic of Korea [2]summary.ai. Correspondence to: Yeongmin Kim <alsdudrla10@kaist.ac.kr>, Il-Chul Moon <icmoon@kaist.ac.kr>.

*Proceedings of the $43^{rd}$ International Conference on Machine Learning*, Seoul, South Korea. PMLR 306, 2026. Copyright 2026 by the author(s).

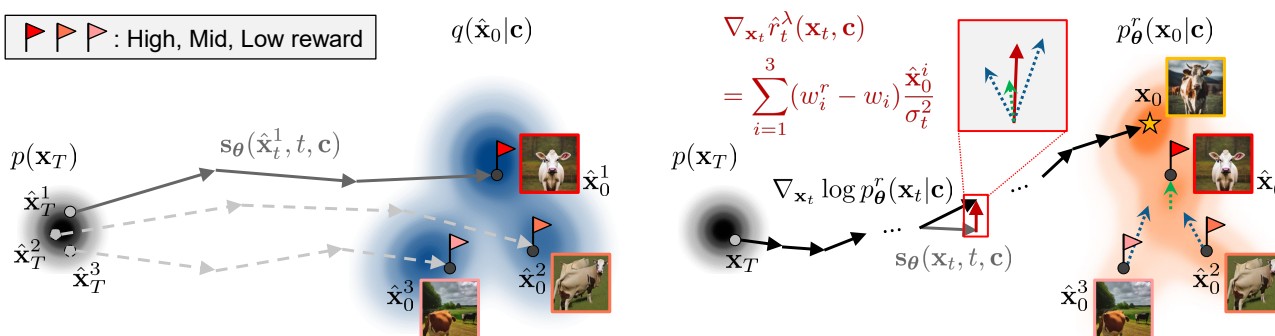

*(a)* Phase 1: Lookahead sampling and reward annotation
*(b)* Phase 2: LiDAR sampling for target reward-tilted distribution

*Figure 1.* Overview of the proposed sampling framework from prompt $\mathbf{c}$. (a) To obtain marginal samples, we generate lookahead samples using a few-step solver and annotate them with a reward function. (b) To generate samples from the target reward-tilted distribution, LiDAR guides particles $\mathbf{x}_t$ toward high-reward lookahead sample $\hat{\mathbf{x}}_0^1$ and repels low-reward lookahead samples $\hat{\mathbf{x}}_0^2, \hat{\mathbf{x}}_0^3$. The guiding weight $w_i^r - w_i$ provided by each lookahead sample $\hat{\mathbf{x}}_0^i$ is proportional to its reward annotation value; see Eq. (16) for the definition.

these approximations remain inaccurate and computationally expensive, as they induce neural dependencies between $\mathbf{x}_t$ and the EFR. In particular, gradient guidance (Bansal et al., 2024; Na et al., 2025) computes the target Stein score, which requires backpropagation through multiple neural networks. To avoid backpropagation through neural networks, approaches based on Sequential Monte Carlo (SMC) have also been proposed (Singhal et al., 2025; Li et al., 2025). However, their performance is highly sensitive to the number of sampling particles of the target distribution, and since importance resampling is done in a high-dimensional image space, the generated samples tend to collapse to nearly identical outcomes (Bengtsson et al., 2008) (See Figure 2).

This paper systematically addresses the previous limitations through a novel approach for approximating the EFR. The proposed formulation detaches the neural dependency between $\mathbf{x}_t$ and the EFR, thereby enabling the computation of the target Stein score in closed form. The proposed EFR formulation is expressed in terms of the final marginal samples of a pre-trained diffusion model and the forward perturbation kernel. Consequently, given a set of pre-generated outputs, both the EFR and the associated Stein score for arbitrary particles can be computed with negligible additional computational cost. The only modest additional cost arises from pre-generating marginal samples for each input prompt; therefore, we propose a lookahead sampling strategy (e.g., a 3-step ODE solver) for efficient marginal sampling, as shown in Figure 1a. We annotate lookahead samples with reward values that guide particles toward high-reward lookahead samples during target sampling, as shown in Figure 1b. We refer to the resulting framework as *Lookahead Sample Reward Guidance* (LiDAR) sampler, and its benefits are summarized in Table 1. LiDAR matches the GenEval performance of the latest gradient guidance method on SDXL while achieving a 9.5× reduction in inference time.

## 2. Preliminaries

### 2.1. Diffusion Models

Diffusion models (Sohl-Dickstein et al., 2015; Ho et al., 2020) define a forward noising process that gradually perturbs a data instance $\mathbf{x}_0 \sim p_{\text{data}}(\mathbf{x}_0)$ into a noisy sample $\mathbf{x}_t$, where the perturbation kernel is given by:

$$p(\mathbf{x}_t|\mathbf{x}_0) = \mathcal{N}(\mathbf{x}_t; \mathbf{x}_0, \sigma_t^2 \mathbf{I}), \tag{1}$$

for all $t \in [0, T]$, where $\sigma_t$ increases monotonically from $\sigma_0 = 0$ to $\sigma_T = \sigma_{\max}$. The gradual perturbation induced by the forward process is also commonly represented by a stochastic differential equation (SDE) (Song et al., 2021b):

$$d\mathbf{x}_t = \mathbf{f}_\sigma(\mathbf{x}_t, t)dt + g_\sigma(t)d\mathbf{w}_t, \tag{2}$$

where $\mathbf{f}_\sigma$ and $g_\sigma$ denote drift and volatility functions corresponding to the noise schedule $\sigma_t$, and $\mathbf{w}_t$ denotes a standard Wiener process. Eq. (2) induces marginal distribution $p(\mathbf{x}_t)$. To generate samples, diffusion models estimate time-dependent *Stein score* $\nabla_{\mathbf{x}_t} \log p(\mathbf{x}_t) \approx \mathbf{s}_\theta(\mathbf{x}_t, t)$, and construct a reverse process (Anderson, 1982) defined as:

$$d\mathbf{x}_t = \left[\mathbf{f}_\sigma(\mathbf{x}_t, t) - g_\sigma^2(t)\mathbf{s}_\theta(\mathbf{x}_t, t)\right]dt + g_\sigma(t)d\mathbf{w}_t. \tag{3}$$

Starting from a prior sample $\mathbf{x}_T \sim p(\mathbf{x}_T)$, the diffusion model generates a sample $\mathbf{x}_0$ by solving Eq. (3).

### 2.2. Test-time Scaling of Diffusion Models

#### 2.2.1. GOAL AND PROBLEM DEFINITION

Test-time scaling (Snell et al., 2025; Muennighoff et al., 2025) methods focus on improving model performance by allocating additional computational resources during inference. A key consideration is how performance gains scale with the added computational cost, including memory usage and inference speed.

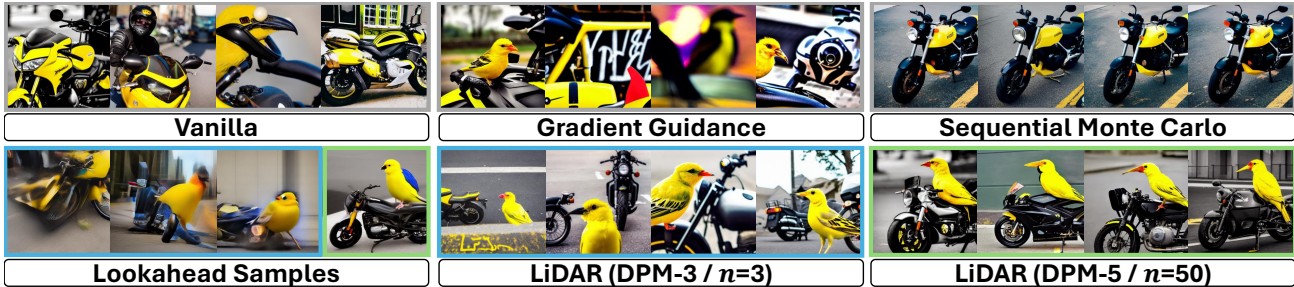

Figure 2. Sampling results for the prompt "*a photo of a yellow bird and a black motorcycle*." Among the lookahead samples, the blue boxes indicate all samples used for LiDAR (DPM-3, $n$=3), ordered from left to right by increasing reward values. The green box indicates the lookahead sample with the highest reward used for LiDAR (DPM-5, $n$=50).

As this paradigm is commonly used in real-world generative services, we primarily focus on text-to-image (T2I) diffusion models (Rombach et al., 2022; Podell et al., 2024), which constitute the most widely adopted application domain of diffusion models. In T2I diffusion models, text $\mathbf{c}$ is used as a conditioning signal for the score network, i.e., $\mathbf{s}_{\boldsymbol{\theta}}(\mathbf{x}_t, t, \mathbf{c}) := \nabla_{\mathbf{x}_t} \log p_{\boldsymbol{\theta}}(\mathbf{x}_t \mid \mathbf{c})$.[1] T2I models generate samples from $p_{\boldsymbol{\theta}}(\mathbf{x}_0|\mathbf{c})$ by solving Eq. (3). We assume access to a pre-trained T2I diffusion model $\mathbf{s}_{\boldsymbol{\theta}}$.

**Problem definition:** The goal of test-time scaling is to generate samples that are better aligned with human intent. Let $r : \mathcal{X}_0 \times \mathcal{C} \to \mathbb{R}$ denote a reward function that reflects human intent, with $\mathcal{X}_0$ being the domain of the final sample $\mathbf{x}_0$ and $\mathcal{C}$ the domain of the text prompt $\mathbf{c}$. The reward function $r$ could be human preference scores or a differentiable neural reward model (Xu et al., 2023a). To achieve this goal, recent approaches (Bansal et al., 2024; Singhal et al., 2025) focus on sampling from a reward-tilted distribution, defined as:

$$p_{\boldsymbol{\theta}}^r(\mathbf{x}_0|\mathbf{c}) \propto p_{\boldsymbol{\theta}}(\mathbf{x}_0|\mathbf{c})\exp\left(\lambda \cdot r(\mathbf{x}_0, \mathbf{c})\right), \qquad (4)$$

where $\lambda$ is a hyperparameter that controls the strength of the reward. To generate samples from the resulting target distribution $p_{\boldsymbol{\theta}}^r$, we need to compute the corresponding time-dependent *target Stein score*:

$$\nabla_{\mathbf{x}_t} \log p_{\boldsymbol{\theta}}^r(\mathbf{x}_t|\mathbf{c}) = \qquad\qquad (5)$$
$$\mathbf{s}_{\boldsymbol{\theta}}(\mathbf{x}_t, t, \mathbf{c}) + \nabla_{\mathbf{x}_t} \underbrace{\log \mathbb{E}_{p_{\boldsymbol{\theta}}(\mathbf{x}_0|\mathbf{x}_t, \mathbf{c})} \left[\exp\left(\lambda \cdot r(\mathbf{x}_0, \mathbf{c})\right)\right]}_{:= r_t^\lambda(\mathbf{x}_t, \mathbf{c}) \text{ (Expected Future Reward)}},$$

where the derivation of Eq. (5) is provided in Section A.1. Given an intermediate particle $\mathbf{x}_t$ during inference, direct computation of the *Expected Future Reward* (EFR) $r_t^\lambda$ is generally intractable; consequently, several approximation methods have been studied.

---

[1]Note that commercial text-to-image diffusion models typically operate in latent space; for simplicity, we denote the diffusion variable as $\mathbf{x}_0$ throughout this paper.

### 2.2.2. COMPUTATION OF EXPECTED FUTURE REWARD

This section describes prior methods for computing the $r_t^\lambda$.

**Backward rollout:** The first approach is to sample from $\mathbf{x}_t$ to $\mathbf{x}_0$ multiple times and estimate the result using the following formula:

$$r_t^\lambda(\mathbf{x}_t, \mathbf{c}) \approx \log \frac{1}{n} \sum_{i=1}^n \exp\left(\lambda \cdot r(\mathbf{x}_0^i, \mathbf{c})\right), \qquad (6)$$

where $\mathbf{x}_0^i \sim p_{\boldsymbol{\theta}}(\mathbf{x}_0|\mathbf{x}_t, \mathbf{c})$. However, each sample $\mathbf{x}_0^i$ requires iterative denoising by solving Eq. (3). Since this procedure must be repeatedly performed for all $t \in [0, T]$, it becomes computationally infeasible. Concurrent works (Holderrieth et al., 2026; Potaptchik et al., 2025; 2026) attempt to reduce computational cost, but rollout sampling still needs to be performed at every timestep.

**Talyer approximation with Tweedie's formula:** This approach employs a first-order Taylor approximation (Chung et al., 2023; Bansal et al., 2024; Na et al., 2025):

$$r_t^\lambda(\mathbf{x}_t, \mathbf{c}) \approx \lambda \cdot r(\bar{\mathbf{x}}_0, \mathbf{c}), \qquad (7)$$

where $\bar{\mathbf{x}}_0 := \mathbb{E}_{p_{\boldsymbol{\theta}}(\mathbf{x}_0|\mathbf{x}_t, \mathbf{c})}[\mathbf{x}_0]$ is given by Tweedie's formula (Robbins, 1992; Efron, 2011). This approximation is relatively practical for test-time scaling, as it requires only a single computation pass through the score network, the decoder (in latent diffusion models), and the reward function $r$ at each $t \in [0, T]$. Both gradient guidance and SMC baselines rely on this approximation. However, rewards are typically well defined only for the iteratively denoised sample $\mathbf{x}_0$, and evaluating them on $\bar{\mathbf{x}}_0$ can be inaccurate, potentially leading to suboptimal performance. As shown in Section A.6 and Figure 8, the Taylor approximation error increases proportionally with $\lambda$; in other words, stronger reward signals lead to larger errors.

### 2.2.3. SAMPLING WITH EXPECTED FUTURE REWARD

This section describes the sampling methods based on the approximated expected future reward in Eq. (7).

**Gradient guidance (Bansal et al., 2024; Na et al., 2025)**: This sampling method approximates the Stein score of $p_{\boldsymbol{\theta}}^r$ in Eq. (5) as follows:

$$\mathbf{s}_{\boldsymbol{\theta}}(\mathbf{x}_t, t, \mathbf{c}) + \lambda \cdot \nabla_{\mathbf{x}_t} r(\bar{\mathbf{x}}_0, \mathbf{c}). \quad (8)$$

This approach has several limitations. First, the inherited approximation of Eq. (7) can introduce errors, leading to inaccurate reward signals. Second, because gradients are taken directly with respect to $\mathbf{x}_t$, the method requires the reward function to be differentiable (i.e., it only supports neural reward models) and is therefore susceptible to reward hacking (Skalse et al., 2022). Finally, a significant drawback is its computational cost: computing $r(\bar{\mathbf{x}}_0, \mathbf{c})$ requires sequentially passing $\mathbf{x}_t$ through the $\mathbf{s}_{\boldsymbol{\theta}}$, the decoder, and the reward model $r$, followed by backpropagation with respect to $\mathbf{x}_t$, making the approach computationally expensive.

**Sequential Monte Carlo (SMC) (Singhal et al., 2025; Li et al., 2025)**: To avoid neural gradient computation, SMC methods reformulate Eq. (5) into a distribution formulation and apply approximation in Eq. (7) as:

$$p_{\boldsymbol{\theta}}^r(\mathbf{x}_t|\mathbf{c}) \propto p_{\boldsymbol{\theta}}(\mathbf{x}_t|\mathbf{c}) \mathbb{E}_{p_{\boldsymbol{\theta}}(\mathbf{x}_0|\mathbf{x}_t, \mathbf{c})} \left[ \exp\left( \lambda \cdot r(\mathbf{x}_0, \mathbf{c}) \right) \right] \quad (9)$$
$$\approx p_{\boldsymbol{\theta}}(\mathbf{x}_t|\mathbf{c}) \exp(\lambda \cdot r(\bar{\mathbf{x}}_0, \mathbf{c})). \quad (10)$$

SMC generates multiple particles $\mathbf{x}_t^i \sim p_{\boldsymbol{\theta}}(\mathbf{x}_t|\mathbf{c})$, for $i \in \{1, \ldots, N\}$, in parallel from a pre-trained diffusion model, and performs importance resampling with weights proportional to $\exp(\lambda \cdot r(\bar{\mathbf{x}}_0, \mathbf{c}))$. Unlike gradient guidance, this approach leverages interactions among multiple particles. Consequently, its performance is highly sensitive to the number of generating particles $N$. Moreover, as shown in Figure 2, the method tends to collapse to a single particle with the highest expected reward, significantly reducing sample diversity, a well-known limitation of particle filtering in high-dimensional data space (Bengtsson et al., 2008).

## 3. Method

Directly solving Eq. (3) with the target Stein score in Eq. (5) constitutes a potentially more promising sampling strategy than the SMC approach in Eq. (9), as it yields consistent performance gains regardless of the number of target samples generated. Accordingly, our primary goal is to overcome the limitations of gradient guidance described in Eq. (8) by proposing a new approximation method for the expected future reward (EFR) $r_t^\lambda$ defined in Eq. (5).

Specifically, Section 3.1 reformulates $r_t^\lambda$ using only future marginal samples and the forward perturbation kernel, which removes the dependence of the intermediate particle $\mathbf{x}_t$ on the neural networks. Section 3.2 introduces an efficient lookahead sampling method for obtaining future marginal samples and presents its weak-to-strong interpretation. Section 3.3 derives a closed-form guidance formulation that avoids backpropagation through neural networks.

---

**Algorithm 1** Lookahead sampling and reward annotation

**Input:** $\mathbf{s}_{\boldsymbol{\theta}}, \mathbf{c}, \delta, n, r$
1. Sample $\hat{\mathbf{x}}_T^i \sim p(\mathbf{x}_T)$ for $i \in \{1, \ldots, n\}$
2. Obtain $\hat{\mathbf{x}}_0^i$ from $\hat{\mathbf{x}}_T^i$ via Eq. (3)
    using $\mathbf{s}_{\boldsymbol{\theta}}(\cdot, \cdot, \mathbf{c})$ with $\delta$ discretization steps
3. $r_i \leftarrow r(\hat{\mathbf{x}}_0^i, \mathbf{c})$ for $i \in \{1, \ldots, n\}$
**Output:** $\{\hat{\mathbf{x}}_0^i, r_i\}_{i=1}^n$

---

**Algorithm 2** LiDAR sampling (Target sampling)

**Input:** $\mathbf{s}_{\boldsymbol{\theta}}, \{\hat{\mathbf{x}}_0^i, r_i\}_{i=1}^n, s, \tau$
1. Sample $\mathbf{x}_T \sim p(\mathbf{x}_T)$
2. $\mathbf{x}_{t_\tau} \leftarrow \mathbf{x}_T$
**for** $k = \{\tau, \ldots, 1\}$ **do:**
    3. Compute $\nabla_{\mathbf{x}_t} \hat{r}_t^\lambda(\mathbf{x}_{t_k}, \mathbf{c})$ via Eq. (16) with $r_i$
    4. Obtain $\mathbf{x}_{t_{k-1}}$ from $\mathbf{x}_{t_k}$ via Eq. (3) using
        $\mathbf{s}_{\boldsymbol{\theta}}(\mathbf{x}_{t_k}, t_k, \mathbf{c}) + s \cdot \nabla_{\mathbf{x}_t} \hat{r}_t^\lambda(\mathbf{x}_{t_k}, \mathbf{c})$
5. $\mathbf{x}_0 \leftarrow \mathbf{x}_{\tau_0}$
**Output:** $\mathbf{x}_0$

---

Finally, Section 3.4 presents scaling laws with respect to lookahead accuracy and the number of lookahead samples.

### 3.1. Expected Future Reward from Future Marginal

The following theorem provides our EFR reformulation:

**Theorem 3.1.** *The expected future reward $r_t^\lambda(\mathbf{x}_t, \mathbf{c})$ defined in Eq.* (5) *can be expressed as*

$$\log \mathbb{E}_{p_{\boldsymbol{\theta}}(\mathbf{x}_0|\mathbf{c})} \left[ \frac{p(\mathbf{x}_t|\mathbf{x}_0)}{\mathbb{E}_{p_{\boldsymbol{\theta}}(\mathbf{x}_0|\mathbf{c})}[p(\mathbf{x}_t|\mathbf{x}_0)]} \exp\left( \lambda \cdot r(\mathbf{x}_0, \mathbf{c}) \right) \right]. \quad (11)$$

Please refer to Section A.2 for the proof. In the previous formulations in Eqs. (6) and (7), the current particle $\mathbf{x}_t$ is fed into neural networks to compute the EFR; consequently, obtaining the Stein score by differentiating with respect to $\mathbf{x}_t$ requires backpropagation through the neural networks.

In contrast, under our formulation in Eq. (11), $\mathbf{x}_t$ is no longer used as an input to the neural models. Previously, $\mathbf{x}_t$ served as an input to $\mathbf{s}_{\boldsymbol{\theta}}$ and subsequently $r$. However, Eq. (11) indicates that $\mathbf{x}_t$ is not used as an input for both $\mathbf{s}_{\boldsymbol{\theta}}$ and $r$ because all samples for the expectation are drawn solely from $\mathbf{c}$, not from $\mathbf{x}_t$. For the expectation computation, the EFR is computed using marginal samples from $p_{\boldsymbol{\theta}}(\mathbf{x}_0|\mathbf{c})$, and the dependence between $\mathbf{x}_t$ and $\mathbf{x}_0$ is mediated through the forward perturbation kernel $p(\mathbf{x}_t|\mathbf{x}_0)$ defined in Eq. (1). We call this rollout method the *forward rollout*.

### 3.2. Efficient Lookahead Sampling for Forward Rollout

Pre-generating samples for arbitrary prompts $\mathbf{c}$ from the pre-trained diffusion model $p_{\boldsymbol{\theta}}(\mathbf{x}_0|\mathbf{c})$ introduces an additional cost in Eq. (11). To further reduce this cost, we alternate sampling from $p_{\boldsymbol{\theta}}(\mathbf{x}_0|\mathbf{c})$ with a faster generator $q(\mathbf{x}_0|\mathbf{c})$, such as a few-step solver (Lu et al., 2022) or a

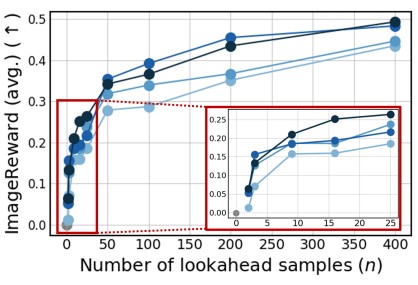 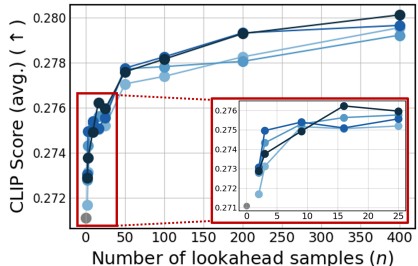 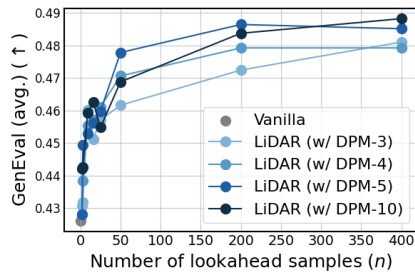

*Figure 3.* Scaling behavior of the LiDAR sampler under different lookahead strategies and varying numbers of lookahead samples $n$. The vanilla method uses only the Stein score $\mathbf{s}_{\boldsymbol{\theta}}$, whereas LiDAR (w/ DPM-$\delta$) incorporates lookahead samples with $\delta$ discretization steps. All results are obtained using SD v1.5 with ImageReward annotations. The legend is shared across all three panels.

distilled model (Song et al., 2023). Algorithm 1 illustrates the lookahead sampling procedure using a $\delta$-step solver.

**Definition 3.2.** We define *lookahead reward* $\tilde{r}_t^\lambda(\mathbf{x}_t, \mathbf{c})$ as:

$$\log \mathbb{E}_{q(\mathbf{x}_0|\mathbf{c})}\left[\frac{p(\mathbf{x}_t|\mathbf{x}_0)}{\mathbb{E}_{q(\mathbf{x}_0|\mathbf{c})}[p(\mathbf{x}_t|\mathbf{x}_0)]}\exp\left(\lambda \cdot r(\mathbf{x}_0, \mathbf{c})\right)\right]. \quad (12)$$

We approximate the target Stein score in Eq. (5) with a *lookahead Stein score*:

$$\mathbf{s}_{\boldsymbol{\theta}}(\mathbf{x}_t, t, \mathbf{c}) + \nabla_{\mathbf{x}_t}\tilde{r}_t^\lambda(\mathbf{x}_t, \mathbf{c}). \quad (13)$$

We denote $\tilde{p}_{\boldsymbol{\theta}}^r(\mathbf{x}_0|\mathbf{c})$ as the resulting distribution by solving Eq. (3) from $\mathbf{x}_T \sim p(\mathbf{x}_T)$, with the lookahead Stein score.

**Weak-to-Strong interpretation:** The proposed lookahead Stein score can be interpreted through the lens of weak-to-strong generalization (Burns et al., 2024; Liu et al., 2024; Zhu et al., 2025), where the reward signal produced by a weak lookahead sampler is transferred to a stronger target sampler. In particular, the proposed lookahead Stein score in Eq. (13) is equivalent (for $s = 1$) to

$$\mathbf{s}_{\boldsymbol{\theta}}(\mathbf{x}_t, t, \mathbf{c}) + s \cdot \nabla_{\mathbf{x}_t}\log\frac{q^r(\mathbf{x}_t \mid \mathbf{c})}{q(\mathbf{x}_t \mid \mathbf{c})}, \quad (14)$$

where $q^r(\mathbf{x}_t|\mathbf{c}) \propto q(\mathbf{x}_t|\mathbf{c})\mathbb{E}_{q(\mathbf{x}_0|\mathbf{x}_t, \mathbf{c})}\left[\exp(\lambda \cdot r(\mathbf{x}_0, \mathbf{c}))\right]$ is defined analogously to Eq. (9), and $q(\mathbf{x}_t|\mathbf{c})$ denotes the marginal distribution induced by $q(\mathbf{x}_0|\mathbf{c})$ and the forward kernel in Eq. (1). From this perspective, adjusting the lookahead reward by a flexible scalar weighting factor $s\ (> 0)$ is commonly adopted (Zhu et al., 2025), and we similarly adopt this weighting to broaden our applicability.

### 3.3. Closed-form Derivative Free Guidance

We compute the lookahead reward in Eq. (12) with finite samples, yielding *empirical lookahead reward* $\hat{r}_t^\lambda(\mathbf{x}_t, \mathbf{c}) :=$

$$\log\frac{1}{n}\sum_{i=1}^{n}\left[\frac{p(\mathbf{x}_t|\hat{\mathbf{x}}_0^i)}{\frac{1}{n}\sum_{j=1}^{n}p(\mathbf{x}_t|\hat{\mathbf{x}}_0^j)}\exp\left(\lambda \cdot r(\hat{\mathbf{x}}_0^i, \mathbf{c})\right)\right], \quad (15)$$

where $\hat{\mathbf{x}}_0^i \sim q(\mathbf{x}_0|\mathbf{c})$ for $i \in \{1, \ldots, n\}$. The previous backward rollout approach in Eq. (6) requires separate samples

for each $t$ and introduces a sequential neural dependency between $\mathbf{x}_t$ and $\mathbf{x}_0$. In contrast, Eq. (15) requires only marginal samples at $t = 0$, and the forward kernel enables computing the EFR for any $\mathbf{x}_t$ without neural dependency between $\mathbf{x}_t$ and $\hat{\mathbf{x}}_0$. As a result, the Stein score of the empirical lookahead reward admits a closed form.

**Theorem 3.3.** *(Derivative-free guidance) Gradient of empirical lookahead reward* $\nabla_{\mathbf{x}_t}\hat{r}_t^\lambda(\mathbf{x}_t, \mathbf{c})$ *has closed-form expression without neural gradient operation as:*

$$\sum_{i=1}^{n}(w_i^r - w_i)\frac{\hat{\mathbf{x}}_0^i}{\sigma_t^2}, \quad (16)$$

*where the weighting functions are:*

$$w_i^r := \text{Softmax}\left(\left\{\lambda \cdot r(\hat{\mathbf{x}}_0^j, \mathbf{c}) - \frac{\|\mathbf{x}_t - \hat{\mathbf{x}}_0^j\|^2}{2\sigma_t^2}\right\}_{j=1}^{n}\right)_i, \quad (17)$$

$$w_i := \text{Softmax}\left(\left\{-\frac{\|\mathbf{x}_t - \hat{\mathbf{x}}_0^j\|^2}{2\sigma_t^2}\right\}_{j=1}^{n}\right)_i. \quad (18)$$

See Section A.3 for the proof. When a lookahead sample $\hat{\mathbf{x}}_0^i$ attains a high reward $r(\hat{\mathbf{x}}_0^i, \mathbf{c})$, its weight $w_i^r$ increases relative to $w_i$, causing the guidance to steer the current particle $\mathbf{x}_t$ toward $\hat{\mathbf{x}}_0^i$, and vice versa. Algorithm 2 provides a procedure for LiDAR sampling with $\tau$ sampling steps, i.e., $(t_\tau = T, t_0 = 0)$.

### 3.4. Probabilistic Scaling Laws

There exists a gap between the proposed generative distribution and the original target distribution $p_{\boldsymbol{\theta}}^r(\mathbf{x}_0|\mathbf{c})$ due to the finite lookahead sampling step $\delta$ and the number of samples $n$. Figure 3 shows clear empirical scaling behavior as $\delta$ and $n$ increase. The following theorems characterize the gap induced by these factors.

**Theorem 3.4.** *(Scaling Law for Lookahead Accuracy $\delta$) Let $\delta$ be the discretization steps for the lookahead sampling, and $\tilde{p}_{\boldsymbol{\theta}}^r(\mathbf{x}_0|\mathbf{c})$ is corresponding resulting distribution with Eq. (13). Then, under mild conditions, the following holds:*

$$D_{TV}(p_{\boldsymbol{\theta}}^r(\mathbf{x}_0|\mathbf{c})||\tilde{p}_{\boldsymbol{\theta}}^r(\mathbf{x}_0|\mathbf{c})) \le \mathcal{O}\left(1/\sqrt{\delta}\right). \quad (19)$$

*Table 2.* Performance on GenEval prompts. ImageReward (IR) is used as the reward for all sampling methods. All performance metrics are averaged over four images per prompt, while Time and Memory report the average cost of generating four images per run on a single A100 GPU. **Bold** values indicate the best results. *: Because generating four images simultaneously causes out-of-memory (OOM) errors, images are generated sequentially with a batch size of 1 over four runs.

| Backbone | Guidance Method | Performance ($\uparrow$) | | | | Inference Cost ($\downarrow$) | |
|---|---|---|---|---|---|---|---|
| | | IR | CLIP | HPS | GenEval | Time (sec.) | Mem. (GiB) |
| SD v1.5 (0.9B) (Rombach et al., 2022) (w/ DDPM 100 steps) | Vanilla | - 0.001 | 0.271 | 0.263 | 0.426 | 7.07 | 8.90 |
| | Vanilla (250-step) | 0.003 | 0.272 | 0.264 | 0.430 | 17.42 | **8.90** |
| | UG (Bansal et al., 2024) | 0.326 | 0.262 | 0.236 | 0.355 | 58.36 | 28.16 |
| | DATE (Na et al., 2025) | 0.364 | 0.274 | 0.267 | 0.438 | 32.89 | 24.71 |
| | LiDAR (DPM-5 / $n$=50) | **0.384** | **0.278** | **0.276** | **0.478** | **13.41** | **8.90** |
| SD v1.5 (0.9B) (Rombach et al., 2022) (w/ DDIM 50 steps) | Vanilla | - 0.125 | 0.269 | 0.270 | 0.423 | 3.58 | 8.90 |
| | Vanilla (200-step) | - 0.112 | 0.269 | 0.270 | 0.415 | 14.09 | **8.90** |
| | UG (Bansal et al., 2024) | 0.201 | 0.259 | 0.236 | 0.344 | 29.59 | 28.16 |
| | DATE (Na et al., 2025) | 0.097 | 0.271 | 0.261 | 0.419 | 17.12 | 24.71 |
| | LiDAR (DPM-5 / $n$=50) | **0.378** | **0.278** | **0.277** | **0.475** | **9.92** | **8.90** |
| SDXL (2.6B) (Podell et al., 2024) (w/ DDPM 100 steps) | Vanilla | 0.722 | 0.282 | 0.292 | 0.545 | 42.00 | 33.84 |
| | Vanilla (250-step) | 0.746 | 0.283 | 0.295 | 0.559 | 104.10 | **33.84** |
| | UG (Bansal et al., 2024) | 0.749 | 0.279 | 0.287 | 0.541 | 334.43 | OOM* |
| | DATE (Na et al., 2025) | 0.960 | 0.283 | 0.294 | 0.570 | 272.32 | OOM* |
| | LiDAR (DPM-8 / $n$=50) | 0.994 | 0.285 | 0.300 | 0.585 | 97.99 | **33.84** |
| | LiDAR (LCM-4 / $n$=100) | **1.007** | **0.286** | 0.300 | 0.585 | 93.18 | **33.84** |
| | LiDAR (DMD-1 / $n$=100) | 1.006 | 0.285 | **0.302** | **0.598** | **78.67** | **33.84** |

Please see Section A.4 for the proof. The resulting upper bound, $\mathcal{O}\left(1/\sqrt{\delta}\right)$, implies that increasing $\delta$ reduces the discretization error of the lookahead sampler, causing it to converge to the target distribution.

**Theorem 3.5.** *(Scaling Law for Lookahead Sample Size $n$) Under mild conditions, for all $\mathbf{x}_t$ and $\mathbf{c}$, the lookahead reward $\tilde{r}_t^\lambda$ and its empirical estimate $\hat{r}_t^\lambda$ satisfy:*

$$\tilde{r}_t^\lambda(\mathbf{x}_t, \mathbf{c}) - \hat{r}_t^\lambda(\mathbf{x}_t, \mathbf{c})$$
$$\xrightarrow{d} \mathcal{N}\left(0, \frac{\mathrm{Var}\left(p(\mathbf{x}_t \mid \mathbf{x}_0) \exp(\lambda \cdot r(\mathbf{x}_0, \mathbf{c}))\right)}{\sqrt{n}\left(\mathbb{E}_{q(\mathbf{x}_0|\mathbf{c})}\left[p(\mathbf{x}_t \mid \mathbf{x}_0)\right] \exp\left(\tilde{r}_t^\lambda(\mathbf{x}_t, \mathbf{c})\right)\right)^2}\right).$$

Please see Section A.5 for the proof. Note that the input to the variance function is a random variable arising from the stochasticity of $\mathbf{x}_0 \sim q(\mathbf{x}_0|\mathbf{c})$. This result shows that using more lookahead samples (i.e., larger $n$) reduces the variance of the approximation to the target lookahead reward. Accordingly, this suggests that the proposed generative distribution approaches $\tilde{p}_{\boldsymbol{\theta}}^r(\mathbf{x}_0|\mathbf{c})$ as $n$ increases.

## 4. Experiments

### 4.1. Comparison to Gradient Guidance Method

This section compares LiDAR against gradient guidance, which serves as our primary baseline. Table 2 compares LiDAR with Universal Guidance (UG) (Bansal et al., 2024) and DATE (Na et al., 2025) using SD v1.5 (Rombach et al.,

2022) and SDXL (Podell et al., 2024) backbones (with CFG (Ho & Salimans, 2021) 7.5). All methods use ImageReward (IR) (Xu et al., 2023a) as the reward function and are evaluated on GenEval (Ghosh et al., 2023) prompts. Following GenEval's protocol, which evaluates four images per prompt based on commercial service use cases (Microsoft Team, 2026; xAI Team, 2026), we generate $N = 4$ images per prompt and report both the average performance and the total cost of generating the four images. We treat the GenEval score as the primary test metric, while using CLIP (Radford et al., 2021) and Human Preference Score (HPS) (Wu et al., 2023b) as validation metrics to mitigate reward hacking in baseline methods.

**Baseline performance**: Table 2 presents an overall comparison of gradient guidance methods. Increasing the number of vanilla sampling steps yields limited improvements across most metrics. In contrast, UG and DATE achieve substantial gains on IR, which is used for guidance. However, UG tends to over-optimize samples for IR, resulting in consistent degradation on other metrics; this trade-off becomes evident as the guidance scale $\lambda$ varies (Figure 4a). DATE addresses this issue by extending UG to update text embeddings and by leveraging information from the $\mathbf{s}_{\boldsymbol{\theta}}$ to update $\mathbf{x}_t$, which improves robustness to reward hacking and allows larger guidance scales that enhance both IR and other metrics. Despite these benefits, gradient guidance methods require approximately $3\times$ more memory than vanilla sampling and incur more than $4\times$ longer runtime in all cases.

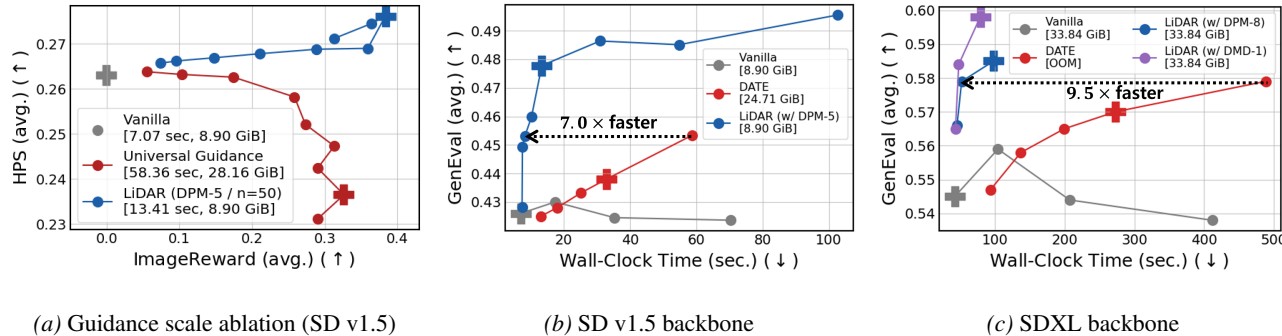

*(a)* Guidance scale ablation (SD v1.5)      *(b)* SD v1.5 backbone      *(c)* SDXL backbone

*Figure 4.* (a) Performance trade-offs between UG and LiDAR. (b, c) Efficiency–performance trade-offs for test-time scaling methods. Vanilla increases sampling steps, DATE adjusts gradient update frequency, and LiDAR controls $n$. The cross marker indicates the performance in Table 2. The black dotted line indicates the time difference for LiDAR to reach DATE's maximum scaling performance.

*Table 3.* Performance of LiDAR when applied to the flow matching model, FLUX (Labs, 2024).

| $n$ | IR | CLIP | GenEval |
|---|---|---|---|
| 0 | 1.019 | 0.282 | 0.645 |
| 3 | 1.038 | 0.282 | 0.663 |
| 50 | 1.114 | 0.283 | **0.668** |
| 100 | **1.198** | **0.284** | 0.667 |

*Table 4.* LiDAR performance on the discrete diffusion model UDLM (Schiff et al., 2025) trained on QM9 (Ruddigkeit et al., 2012), using the ring count reward.

| $n$ | Num novel ($\uparrow$) | Novel ring count ($\uparrow$) | Total ring count ($\uparrow$) |
|---|---|---|---|
| 0 | 130 | 2.192 | 1.753 |
| 16 | 205 | 3.034 | 2.601 |
| 1024 | 251 | 3.948 | **3.393** |
| 4096 | **257** | **4.128** | 3.067 |

**LiDAR against baselines**: The proposed LiDAR in Table 2 outperforms all baselines in both performance and inference cost across all backbones and target sampler types. LiDAR uses a DPM solver (Lu et al., 2022) as the default lookahead solver, where DPM-5 indicates $\delta = 5$. When using $n = 50$ lookahead samples, the memory usage and runtime of the target sampling (Algorithm 2) stage are nearly identical to those of vanilla sampling (See Figure 10). The additional cost relative to vanilla sampling arises solely from the lookahead sampling and reward annotation stages (Algorithm 1). Please refer to Table 9 for the cost of each component. Because LiDAR achieves strong performance with relatively small $n$, the overall computational overhead remains modest, making it more efficient than gradient guidance.

**Efficiency-performance control curve**: Figures 4b and 4c compare efficiency-performance trade-offs as the scaling factor of each method varies, showing that LiDAR consistently envelopes the baselines. While DATE cannot exploit additional computation beyond its peak performance, which is achieved by applying gradient updates at every step, LiDAR continues to scale with larger $n$ or by improving lookahead accuracy. In particular, LiDAR achieves the peak SDXL performance of DATE approximately 9.5× faster.

**Leveraging distillation models**: We also evaluate lookahead strategies using distillation models, such as LCM-LoRA (4 steps) (Luo et al., 2023) and DMD (1 step) (Yin et al., 2024), as shown in the SDXL block of Table 2. These models provide more accurate lookahead samples per step, resulting in improved efficiency-performance trade-offs.

### 4.2. Analysis

**Extension to flow matching models:** Since flow matching models (Lipman et al., 2023) can be formulated equivalently to diffusion models (Gao et al., 2024), LiDAR can be applied straightforwardly. As shown in Table 3, performance on FLUX (Labs, 2024) with the IR reward function improves as the number of lookahead samples increases. We use Hyper-Flux-8 (Ren et al., 2024) as the lookahead sampler.

**Extension to discrete diffusion models:** For discrete diffusion models, although the Stein score formulation is unavailable, the lookahead reward formulation in Eq. (12) remains applicable, as detailed in Section A.7. Following Discrete Classifier Guidance (Schiff et al., 2025), we assume dimension-wise independence when constructing the guidance term. As shown in Table 4, LiDAR can be successfully extended to discrete diffusion models, with performance improving as the number of lookahead samples increases. However, a limitation of our approach is that the sequence-level reward guidance is incorporated in a token-independent manner. Developing more expressive guidance mechanisms that capture token dependencies is an important direction for future work. We use an 8-step solver for lookahead sampling and a 32-step solver for target sampling.

*Table 5.* Orthogonal improvements applied to a DPO-tuned SD v1.5 (Wallace et al., 2024). *: Fine-tuning costs are not included.

| Lookahead | $n$ | IR | CLIP | GenEval | Time |
|---|---|---|---|---|---|
| - | 0 | - 0.001 | 0.271 | 0.426 | 7.07 |
| DPM-3 | 3 | 0.109 | 0.274 | 0.439 | 7.44 |
| DPM-5 | 3 | 0.172 | 0.275 | 0.449 | 7.54 |
| DPM-5 | 9 | 0.211 | 0.276 | 0.453 | 8.27 |
| DPM-5 | 50 | **0.384** | **0.278** | **0.478** | 13.41 |
| (+ DPO) | 0 | 0.106 | 0.274 | 0.446 | 7.07* |
| DPM-5 | 3 | 0.268 | 0.277 | 0.452 | 7.54* |
| DPM-5 | 9 | 0.296 | 0.277 | 0.468 | 8.27* |
| DPM-5 | 50 | **0.445** | **0.280** | **0.489** | 13.41* |

*Table 6.* Performance on SD v1.5 using a weighted sum of ImageReward and CLIP as reward annotations. All results use a DPM-5 lookahead solver with $n = 50$. **Bold** values indicate the best results.

| Reward Mix | | Performance ($\uparrow$) | | | |
|---|---|---|---|---|---|
| IR | CLIP | IR | CLIP | HPS | GenEval |
| 0 % | 0 % | - 0.001 | 0.2711 | 0.263 | 0.426 |
| 0 % | 100 % | 0.213 | 0.2799 | 0.264 | 0.454 |
| 10 % | 90 % | 0.278 | **0.2801** | 0.265 | 0.459 |
| 40 % | 60 % | 0.374 | 0.2799 | 0.268 | 0.476 |
| 90 % | 10 % | **0.392** | 0.2781 | 0.268 | 0.473 |
| 100 % | 0 % | 0.384 | 0.2781 | **0.276** | **0.478** |

*Table 7.* Performance comparison with other sample-based guidance methods using our lookahead samples based on DPM-5 lookahead solver with $n = 50$.

| Method | IR | CLIP | HPS |
|---|---|---|---|
| Vanilla | - 0.001 | 0.271 | 0.263 |
| Safe-D (Kim et al., 2025a) | - 0.001 | 0.271 | 0.262 |
| SR (Kirchhof et al., 2025) | 0.014 | 0.272 | 0.263 |
| LiDAR | **0.384** | **0.278** | **0.276** |

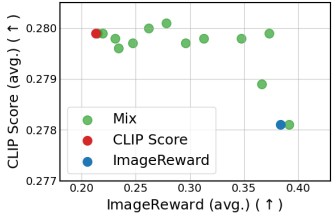

*Figure 5.* Results on mixed reward annotations on lookahead samples.

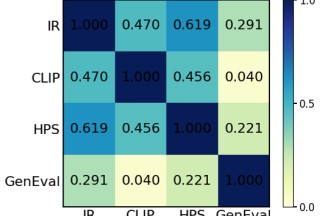

*Figure 6.* Pairwise correlations between metrics on lookahead samples (DPM-5).

**Performance gains with extremely small additional cost:** Using only a small number of lookahead samples (e.g., $n = 3$ with a DPM-3 lookahead solver) yields meaningful performance gains, as shown in Table 5 and highlighted by the red boxes in Figure 3. Figure 2 presents qualitative examples of lookahead samples alongside their corresponding target outputs. Although all lookahead samples are visually coarse, some are clearly better than others, and providing a signal that favors better lookahead samples over worse ones appears to benefit the target LiDAR sampler.

**Orthogonal to fine-tuning method:** Table 5 shows that applying LiDAR to a DPO-tuned SD v1.5 (Wallace et al., 2024) yields additional performance gains that are orthogonal to fine-tuning. Notably, the improvements from LiDAR are larger than those achieved by fine-tuning alone.

**Why have all the metrics improved in LiDAR?:** Although LiDAR uses IR to annotate lookahead samples, it also improves other metrics, unlike UG. Figure 6 shows that IR is correlated with other metrics for lookahead samples; therefore, guiding target particles toward high-IR lookahead samples naturally steers sampling in directions that improve multiple metrics.

**Performance with other reward function:** Table 6 shows results using CLIP or a weighted combination of IR and CLIP as the reward. Increasing the weight of IR improves IR scores, while increasing the weight of CLIP improves CLIP scores. Incorporating a small fraction of the other reward

function (10%) yields slightly better performance than using a single metric alone, likely due to the correlations shown in Figure 6. The IR reward function achieves higher HPS and GenEval scores than the CLIP reward function, reflecting stronger correlations with these metrics. The combination of IR and CLIP reward function produces frontier points for these metrics, as illustrated in Figure 5.

**Comparison to other sample-based guidance:** Sample-based diffusion guidance has been explored for safe generation (Kim et al., 2025a; Kirchhof et al., 2025), typically assuming access to negative samples. We treat low-reward lookahead samples as negative samples and apply existing guidance methods to evaluate this setting. As shown in Table 7, negative guidance is ineffective for reward alignment. In Safe-D (Kim et al., 2025a), guidance is applied only at diffusion timesteps close to pure noise, which is effective for safety tasks where safe and unsafe samples differ at a coarse semantic level. In contrast, image–text alignment requires fine-grained control over attributes such as color and spatial configuration, rendering simple avoidance of undesirable samples insufficient. LiDAR leverages soft reward values to both repel low-quality samples and attract high-quality ones, enabling effective incorporation of reward information.

**Ablation on $\lambda$ and approximation quality on EFR:** The left panel of Figure 8 presents an ablation study on $\lambda$, showing that increasing $\lambda$ consistently improves performance across all metrics. We therefore fix $\lambda = 5000$ for all subsequent experiments. Although this choice slightly reduces

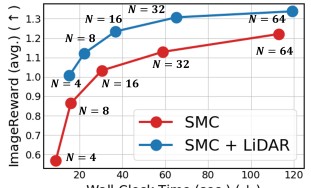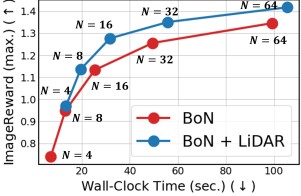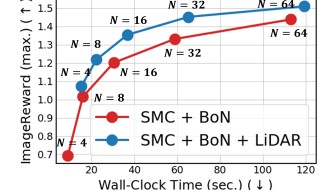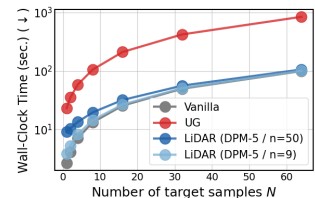

*Figure 7.* (Left three) Efficiency–performance trade-offs of LiDAR applied to each method as the number of target samples $N$ varies. (Right) Time required to generate $N$ samples.

diversity, LiDAR continues to produce meaningfully diverse samples, as shown in Figure 2. In contrast, larger $\lambda$ leads to severe performance degradation in UG (See Figure 4a for $\lambda$ scaling in UG; for LiDAR, we fix $\lambda$ and adjust $s$), as samples are pushed away from the data manifold, resulting in unnatural outputs (See Figure 2). This behavior arises from UG's Taylor approximation, whose error scales proportionally with $\lambda$ (See Section A.6). The right panel of Figure 8 reports the mean squared error between the approximated and target EFR as a function of $\lambda$, demonstrating that LiDAR achieves substantially more accurate EFR estimates.

### 4.3. Orthogonal Integration with SMC and Best-of-$N$

Sequential Monte Carlo (SMC) (Singhal et al., 2025) and Best-of-$N$ (BoN) are not comparable to the setting in Section 4.1, where four samples are generated and presented simultaneously. The performance of SMC depends on the number of generated target samples $N$. When only four samples are generated, the outputs are nearly identical and unsuitable for providing multiple samples (See Figure 2). Meanwhile, BoN generates $N$ samples but returns only a single one, resulting in a fundamentally different use case.

These use cases are practically meaningful in certain aspects, so we further combine each method with LiDAR in an orthogonal manner. As shown in Table 8, applying LiDAR to SMC, BoN, and SMC (+ BoN) improves performance. Although LiDAR requires lookahead sampling, Figure 7 shows that it achieves better efficiency–performance trade-offs. LiDAR becomes more efficient as $N$ increases because the $n$ lookahead samples could be shared across the $N$ target samples. The sampling time converges to that of the vanilla sampling for large $N$, as shown in the right panel of Figure 7. This explains why LiDAR is well-suited to being combined with methods such as SMC and BoN, which generate a large number of samples from the outset.

## 5. Conclusion

We propose LiDAR, a test-time scaling method for diffusion models that consists of lookahead sampling with reward-guided target sampling. LiDAR systematically addresses the efficiency and reward-hacking limitations of gradient

*Table 8.* ImageReward performance with LiDAR (DPM-5, $n = 50$) orthogonally applied to target particle interaction methods. Since performance varies with the number of generated samples $N$ on these methods, we report results for multiple values of $N$.

| Method | 1 | 2 | 4 | 8 | 16 | 32 | 64 |
|---|---|---|---|---|---|---|---|
| SMC | - 0.001 | 0.267 | 0.568 | 0.864 | 1.033 | 1.129 | 1.221 |
| + LiDAR | **0.384** | **0.715** | **1.007** | **1.122** | **1.234** | **1.307** | **1.338** |
| BoN | - 0.001 | 0.414 | 0.739 | 0.948 | 1.133 | 1.256 | 1.346 |
| + LiDAR | **0.384** | **0.749** | **0.969** | **1.136** | **1.277** | **1.351** | **1.418** |
| SMC (+ BoN) | - 0.001 | 0.336 | 0.692 | 1.015 | 1.201 | 1.331 | 1.438 |
| + LiDAR | **0.384** | **0.770** | **1.073** | **1.218** | **1.354** | **1.452** | **1.510** |

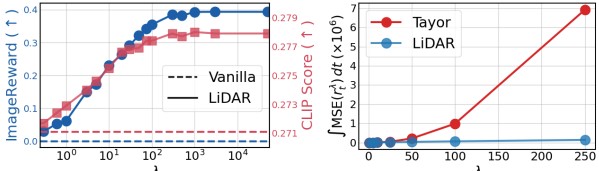

*Figure 8.* (Left) Ablation on $\lambda$ for LiDAR. (Right) EFR approximation error of Taylor and LiDAR according to $\lambda$.

guidance methods. By decoupling the neural dependency between the expected future reward and $\mathbf{x}_t$, LiDAR enables derivative-free guidance without relying on SMC. LiDAR exhibits strong scaling behavior with respect to both lookahead accuracy and the number of lookahead samples, achieving significantly improved efficiency–performance trade-offs compared to existing test-time scaling methods.

## Impact Statement

The primary goal of this work is to improve human alignment in the diffusion sampling process. The proposed method may have potential applications in creative domains such as digital media, visual content generation, and virtual environments. At the same time, it is important to consider the ethical use of AI-generated content and the risk of producing misleading or harmful material. Care should be taken to avoid applying the proposed approach in conjunction with harmful or poorly designed reward functions. Possible mitigation strategies include integrating protection mechanisms and invisible watermarking to reduce potential misuse.

# Acknowledgements

This work was supported by the IITP (Institute of Information & Communications Technology Planning & Evaluation)-ITRC (Information Technology Research Center) grant funded by the Korea government (Ministry of Science and ICT) (IITP-2026-RS-2024-00437268) (50%). This research was supported by AI Technology Development for Commonsense Extraction, Reasoning, and Inference from Heterogeneous Data(IITP) funded by the Ministry of Science and ICT(RS-2022-II220077) (50%).

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

# A. Proofs and Derivations

## A.1. Derivation of Intermediate Stein Score in Eq. (5)

We have followings in Eq. (4):

$$p_\theta^r(\mathbf{x}_0|\mathbf{c}) \propto p_\theta(\mathbf{x}_0|\mathbf{c})\exp\left(\lambda \cdot r(\mathbf{x}_0, \mathbf{c})\right).$$

By marginalizing over $\mathbf{x}_0$, the marginal distribution at time $t$ is

$$p_\theta^r(\mathbf{x}_t|\mathbf{c}) = \int p_\theta^r(\mathbf{x}_0|\mathbf{c})\, p(\mathbf{x}_t|\mathbf{x}_0)\, d\mathbf{x}_0$$
$$\propto \int p_\theta(\mathbf{x}_0|\mathbf{c})\exp\left(\lambda \cdot r(\mathbf{x}_0, \mathbf{c})\right)\, p(\mathbf{x}_t|\mathbf{x}_0)\, d\mathbf{x}_0.$$

Using Bayes' rule and the definition of a forward process,

$$p(\mathbf{x}_t|\mathbf{x}_0) = p(\mathbf{x}_t|\mathbf{x}_0, \mathbf{c}) = \frac{p_\theta(\mathbf{x}_0|\mathbf{x}_t, \mathbf{c})\, p_\theta(\mathbf{x}_t|\mathbf{c})}{p_\theta(\mathbf{x}_0|\mathbf{c})},$$

we obtain

$$p_\theta^r(\mathbf{x}_t|\mathbf{c}) \propto \int p_\theta(\mathbf{x}_0|\mathbf{c})\exp\left(\lambda \cdot r(\mathbf{x}_0, \mathbf{c})\right) \frac{p_\theta(\mathbf{x}_0|\mathbf{x}_t, \mathbf{c})\, p_\theta(\mathbf{x}_t|\mathbf{c})}{p_\theta(\mathbf{x}_0|\mathbf{c})}\, d\mathbf{x}_0$$
$$= p_\theta(\mathbf{x}_t|\mathbf{c}) \int \exp\left(\lambda \cdot r(\mathbf{x}_0, \mathbf{c})\right) p_\theta(\mathbf{x}_0|\mathbf{x}_t, \mathbf{c})\, d\mathbf{x}_0$$
$$= p_\theta(\mathbf{x}_t|\mathbf{c})\, \mathbb{E}_{p_\theta(\mathbf{x}_0|\mathbf{x}_t, \mathbf{c})}\left[\exp\left(\lambda \cdot r(\mathbf{x}_0, \mathbf{c})\right)\right].$$

Taking the gradient of the log-density w.r.t to $\mathbf{x}_t$, we have

$$\nabla_{\mathbf{x}_t} \log p_\theta^r(\mathbf{x}_t|\mathbf{c}) = \nabla_{\mathbf{x}_t} \log p_\theta(\mathbf{x}_t|\mathbf{c}) + \nabla_{\mathbf{x}_t} \log \mathbb{E}_{p_\theta(\mathbf{x}_0|\mathbf{x}_t, \mathbf{c})}\left[\exp\left(\lambda \cdot r(\mathbf{x}_0, \mathbf{c})\right)\right]$$
$$= \mathbf{s}_\theta(\mathbf{x}_t, t, \mathbf{c}) + \nabla_{\mathbf{x}_t} \log \mathbb{E}_{p_\theta(\mathbf{x}_0|\mathbf{x}_t, \mathbf{c})}\left[\exp\left(\lambda \cdot r(\mathbf{x}_0, \mathbf{c})\right)\right].$$

## A.2. Proof of Theorem 3.1

**Theorem 3.1.** *The expected future reward $r_t^\lambda(\mathbf{x}_t, \mathbf{c})$ defined in Eq. (5) can be expressed as*

$$\log \mathbb{E}_{p_\theta(\mathbf{x}_0|\mathbf{c})}\left[\frac{p(\mathbf{x}_t|\mathbf{x}_0)}{\mathbb{E}_{p_\theta(\mathbf{x}_0|\mathbf{c})}[p(\mathbf{x}_t|\mathbf{x}_0)]}\exp\left(\lambda \cdot r(\mathbf{x}_0, \mathbf{c})\right)\right]. \tag{11}$$

*Proof.* We have

$$r_t^\lambda(\mathbf{x}_t, \mathbf{c}) = \log \mathbb{E}_{p_\theta(\mathbf{x}_0|\mathbf{x}_t, \mathbf{c})}\left[\exp\left(\lambda \cdot r(\mathbf{x}_0, \mathbf{c})\right)\right]$$
$$= \log \int p_\theta(\mathbf{x}_0|\mathbf{x}_t, \mathbf{c})\exp\left(\lambda \cdot r(\mathbf{x}_0, \mathbf{c})\right)\, d\mathbf{x}_0$$
$$= \log \int \frac{p_\theta(\mathbf{x}_0, \mathbf{x}_t|\mathbf{c})}{p_\theta(\mathbf{x}_t|\mathbf{c})}\exp\left(\lambda \cdot r(\mathbf{x}_0, \mathbf{c})\right)\, d\mathbf{x}_0$$
$$= \log \int \frac{p(\mathbf{x}_t|\mathbf{x}_0)\, p_\theta(\mathbf{x}_0|\mathbf{c})}{\int p(\mathbf{x}_t|\mathbf{x}_0)\, p_\theta(\mathbf{x}_0|\mathbf{c})\, d\mathbf{x}_0}\exp\left(\lambda \cdot r(\mathbf{x}_0, \mathbf{c})\right)\, d\mathbf{x}_0$$
$$= \log \mathbb{E}_{p_\theta(\mathbf{x}_0|\mathbf{c})}\left[\frac{p(\mathbf{x}_t|\mathbf{x}_0)}{\mathbb{E}_{p_\theta(\mathbf{x}_0|\mathbf{c})}[p(\mathbf{x}_t|\mathbf{x}_0)]}\exp\left(\lambda \cdot r(\mathbf{x}_0, \mathbf{c})\right)\right].$$

$\square$

## A.3. Proof of Theorem 3.3

**Theorem 3.3.** *(Derivative-free guidance) Gradient of empirical lookahead reward $\nabla_{\mathbf{x}_t} \hat{r}_t^\lambda(\mathbf{x}_t, \mathbf{c})$ has closed-form expression without neural gradient operation as:*

$$\sum_{i=1}^n (w_i^r - w_i)\frac{\hat{\mathbf{x}}_0^i}{\sigma_t^2}, \tag{16}$$

*where the weighting functions are:*

$$w_i^r := \mathrm{Softmax}\left(\left\{\lambda \cdot r(\hat{\mathbf{x}}_0^j, \mathbf{c}) - \frac{\|\mathbf{x}_t - \hat{\mathbf{x}}_0^j\|^2}{2\sigma_t^2}\right\}_{j=1}^n\right)_i, \tag{17}$$

$$w_i := \mathrm{Softmax}\left(\left\{-\frac{\|\mathbf{x}_t - \hat{\mathbf{x}}_0^j\|^2}{2\sigma_t^2}\right\}_{j=1}^n\right)_i. \tag{18}$$

*Proof.* We have a forward Gaussian kernel and its Stein score as:

$$p(\mathbf{x}_t|\mathbf{x}_0) = \frac{1}{\sigma_t\sqrt{2\pi}}\exp\left(-\frac{(\mathbf{x}_t - \mathbf{x}_0)^2}{2\sigma_t^2}\right), \qquad \nabla_{\mathbf{x}_t}\log p(\mathbf{x}_t|\mathbf{x}_0) = -\frac{\mathbf{x}_t - \mathbf{x}_0}{\sigma_t^2}.$$

The definition of empirical lookahead reward is

$$\hat{r}_t^\lambda(\mathbf{x}_t, \mathbf{c}) := \log \frac{1}{n}\sum_{i=1}^n \left[\frac{p(\mathbf{x}_t|\hat{\mathbf{x}}_0^i)}{\frac{1}{n}\sum_{j=1}^n p(\mathbf{x}_t|\hat{\mathbf{x}}_0^j)}\exp\left(\lambda \cdot r(\hat{\mathbf{x}}_0^i, \mathbf{c})\right)\right].$$

$$\nabla_{\mathbf{x}_t} \hat{r}_t^\lambda(\mathbf{x}_t, \mathbf{c})$$

$$= \nabla_{\mathbf{x}_t} \log \frac{1}{n} \sum_{i=1}^n \frac{p(\mathbf{x}_t|\hat{\mathbf{x}}_0^i)}{\frac{1}{n}\sum_{j=1}^n p(\mathbf{x}_t|\hat{\mathbf{x}}_0^j)} \exp(\lambda \cdot r(\hat{\mathbf{x}}_0^i, \mathbf{c}))$$

$$= \nabla_{\mathbf{x}_t} \log \left[ \frac{1}{n} \sum_{i=1}^n \frac{\exp\left(-\frac{(\mathbf{x}_t-\hat{\mathbf{x}}_0^i)^2}{2\sigma_t^2}\right)}{\frac{1}{n}\sum_{j=1}^n \exp\left(-\frac{(\mathbf{x}_t-\hat{\mathbf{x}}_0^j)^2}{2\sigma_t^2}\right)} \exp(\lambda \cdot r(\hat{\mathbf{x}}_0^i, \mathbf{c})) \right]$$

$$= \nabla_{\mathbf{x}_t} \log \left[ \sum_{i=1}^n \exp\left(-\frac{(\mathbf{x}_t-\hat{\mathbf{x}}_0^i)^2}{2\sigma_t^2}\right)\exp(\lambda \cdot r(\hat{\mathbf{x}}_0^i, \mathbf{c})) \right] - \nabla_{\mathbf{x}_t} \log \left[ \sum_{j=1}^n \exp\left(-\frac{(\mathbf{x}_t-\hat{\mathbf{x}}_0^j)^2}{2\sigma_t^2}\right) \right]$$

$$= \frac{\sum_{i=1}^n \nabla_{\mathbf{x}_t} \exp\left((\lambda \cdot r(\hat{\mathbf{x}}_0^i, \mathbf{c})) - \frac{(\mathbf{x}_t-\hat{\mathbf{x}}_0^i)^2}{2\sigma_t^2}\right)}{\sum_{j=1}^n \exp\left((\lambda \cdot r(\hat{\mathbf{x}}_0^j, \mathbf{c})) - \frac{(\mathbf{x}_t-\hat{\mathbf{x}}_0^j)^2}{2\sigma_t^2}\right)} - \frac{\sum_{i=1}^n \nabla_{\mathbf{x}_t} \exp\left(-\frac{(\mathbf{x}_t-\hat{\mathbf{x}}_0^i)^2}{2\sigma_t^2}\right)}{\sum_{j=1}^n \exp\left(-\frac{(\mathbf{x}_t-\hat{\mathbf{x}}_0^j)^2}{2\sigma_t^2}\right)}$$

$$= \frac{\sum_{i=1}^n \exp\left((\lambda \cdot r(\hat{\mathbf{x}}_0^i, \mathbf{c})) - \frac{(\mathbf{x}_t-\hat{\mathbf{x}}_0^i)^2}{2\sigma_t^2}\right) \nabla_{\mathbf{x}_t}\left((\lambda \cdot r(\hat{\mathbf{x}}_0^i, \mathbf{c})) - \frac{(\mathbf{x}_t-\hat{\mathbf{x}}_0^i)^2}{2\sigma_t^2}\right)}{\sum_{j=1}^n \exp\left((\lambda \cdot r(\hat{\mathbf{x}}_0^j, \mathbf{c})) - \frac{(\mathbf{x}_t-\hat{\mathbf{x}}_0^j)^2}{2\sigma_t^2}\right)}$$

$$\quad - \frac{\sum_{i=1}^n \exp\left(-\frac{(\mathbf{x}_t-\hat{\mathbf{x}}_0^i)^2}{2\sigma_t^2}\right) \nabla_{\mathbf{x}_t}\left(-\frac{(\mathbf{x}_t-\hat{\mathbf{x}}_0^i)^2}{2\sigma_t^2}\right)}{\sum_{j=1}^n \exp\left(-\frac{(\mathbf{x}_t-\hat{\mathbf{x}}_0^j)^2}{2\sigma_t^2}\right)}$$

$$= \sum_{i=1}^n w_i^r \nabla_{\mathbf{x}_t}\left((\lambda \cdot r(\hat{\mathbf{x}}_0^i, \mathbf{c})) - \frac{(\mathbf{x}_t-\hat{\mathbf{x}}_0^i)^2}{2\sigma_t^2}\right) - \sum_{i=1}^n w_i \nabla_{\mathbf{x}_t}\left(-\frac{(\mathbf{x}_t-\hat{\mathbf{x}}_0^i)^2}{2\sigma_t^2}\right)$$

$$= \sum_{i=1}^n w_i^r \nabla_{\mathbf{x}_t}\left(-\frac{(\mathbf{x}_t-\hat{\mathbf{x}}_0^i)^2}{2\sigma_t^2}\right) - \sum_{i=1}^n w_i \nabla_{\mathbf{x}_t}\left(-\frac{(\mathbf{x}_t-\hat{\mathbf{x}}_0^i)^2}{2\sigma_t^2}\right)$$

$$= \sum_{i=1}^n (w_i^r - w_i) \nabla_{\mathbf{x}_t}\left(-\frac{(\mathbf{x}_t-\hat{\mathbf{x}}_0^i)^2}{2\sigma_t^2}\right)$$

$$= \sum_{i=1}^n (w_i^r - w_i)\left(\frac{1}{\sigma_t^2}\right)(\hat{\mathbf{x}}_0^i - \mathbf{x}_t)$$

$$= \sum_{i=1}^n (w_i^r - w_i)\left(\frac{1}{\sigma_t^2}\right)(\hat{\mathbf{x}}_0^i)$$

$\square$

## A.4. Proof of Theorem 3.4

**Theorem 3.4.** *(Scaling Law for Lookahead Accuracy $\delta$) Let $\delta$ be the discretization steps for the lookahead sampling, and $\tilde{p}_{\boldsymbol{\theta}}^r(\mathbf{x}_0|\mathbf{c})$ is corresponding resulting distribution with Eq. (13). Then, under mild conditions, the following holds:*

$$D_{TV}(p_{\boldsymbol{\theta}}^r(\mathbf{x}_0|\mathbf{c})||\tilde{p}_{\boldsymbol{\theta}}^r(\mathbf{x}_0|\mathbf{c})) \leq \mathcal{O}\left(1/\sqrt{\delta}\right). \tag{19}$$

*Proof.* We define the reward-tilted distribution as $q^r(\mathbf{x}_0|\mathbf{c}) \propto q(\mathbf{x}_0|\mathbf{c})\exp\left(\lambda \cdot r(\mathbf{x}_0, \mathbf{c})\right)$. Similar with Section A.1, we have the reward-tilted marginal distribution as follows: $q^r(\mathbf{x}_t|\mathbf{c}) \propto q(\mathbf{x}_t|\mathbf{c})\mathbb{E}_{q(\mathbf{x}_0|\mathbf{x}_t,\mathbf{c})}\left[\exp\left(\lambda \cdot r(\mathbf{x}_0, \mathbf{c})\right)\right]$. Let the time partition be $0 = t_1 < t_2 < \cdots < t_\delta = T$. $\{\mathbf{x}_{t_k}\}_{k=0}^\delta$ denotes the set of the Euler-Maruyama method with $q$ i.e., $\mathbf{x}_{t_k} = \mathbf{x}_{t_{k+1}} + \left(f_\sigma(\mathbf{x}_{t_{k+1}}, t_{k+1}) - g_\sigma^2(t_{k+1})\nabla_{\mathbf{x}_{t_{k+1}}} \log q(\mathbf{x}_{t_{k+1}})\right)\delta + g_\sigma(t_{k+1})\sqrt{\delta} \cdot Z_{k+1}$ where $Z_{k+1} \sim \mathcal{N}(Z_{k+1}; 0, \mathbf{I})$. Also, let $\{\hat{\mathbf{x}}_t\}_{t\in[0,T]}$ be a continuous-time process from $\{\mathbf{x}_{t_k}\}_{k=0}^\delta$ where $\hat{\mathbf{x}}_t := \mathbf{x}_{t_k}$ for $t_k \leq t < t_{k+1}$.

We make the following assumptions, some of which are widely used (Song & Ermon, 2019; Song et al., 2021a):

1. $f_\sigma(\mathbf{x}_t, t)$ and $g_\sigma(t)$ satisfy Lipschitz continuity and linear growth conditions.

2. $\exists g_{max}, \forall t : g_\sigma(t) \leq g_{\max}$.

3. $\exists L, \forall \mathbf{x}, \mathbf{y}, \mathbf{c} : \|\nabla_{\mathbf{x}} \log p_{\boldsymbol{\theta}}(\mathbf{x}|\mathbf{c}) - \nabla_{\mathbf{y}} \log p_{\boldsymbol{\theta}}(\mathbf{y}|\mathbf{c})\|_2 \leq L \cdot \|\mathbf{x} - \mathbf{y}\|_2$.

4. $\exists r_{min}, r_{max}, \forall \mathbf{x}_0, \mathbf{c} : r_{min} \leq r(\mathbf{x}_0, \mathbf{c}) \leq r_{max}$.

5. Assumptions of Theorem 4 in Kim et al. (2022).

6. $\forall k, \mathbf{c} : \nabla_{t_k} \log p_{\boldsymbol{\theta}}(\mathbf{x}_{t_k}|\mathbf{c}) = \nabla_{t_k} \log q(\mathbf{x}_{t_k}|\mathbf{c})$.

Furthermore, we define some notations for brevity as follows: $w(\mathbf{x}_0, \mathbf{c}) := \exp(\lambda \cdot r(\mathbf{x}_0, \mathbf{c}))$, $m := \exp(\lambda \cdot r_{min})$, $M := \exp(\lambda \cdot r_{max})$, $Z_p(\mathbf{c}) := \int p_{\boldsymbol{\theta}}(\mathbf{x}_0|\mathbf{c})w(\mathbf{x}_0, \mathbf{c})d\mathbf{x}_0$, and $Z_q(\mathbf{c}) := \int q(\mathbf{x}_0|\mathbf{c})w(\mathbf{x}_0, \mathbf{c})d\mathbf{x}_0$. Note that $w(\mathbf{x}_0, \mathbf{c}), m, M, Z_p(\mathbf{c}), Z_q(\mathbf{c}) \geq 0$.

We begin the proof using $q^r(\mathbf{x}_0|\mathbf{c})$ and the triangle inequality for the upper bound of the lookahead error:

$$D_{TV}(p_{\boldsymbol{\theta}}^r(\mathbf{x}_0|\mathbf{c})||\tilde{p}_{\boldsymbol{\theta}}^r(\mathbf{x}_0|\mathbf{c})) \leq D_{TV}(p_{\boldsymbol{\theta}}^r(\mathbf{x}_0|\mathbf{c})||q^r(\mathbf{x}_0|\mathbf{c})) + D_{TV}(q^r(\mathbf{x}_0|\mathbf{c})||\tilde{p}_{\boldsymbol{\theta}}^r(\mathbf{x}_0|\mathbf{c})) \tag{20}$$

$$= D_{TV}(p_{\boldsymbol{\theta}}^r(\mathbf{x}_0|\mathbf{c})||q^r(\mathbf{x}_0|\mathbf{c})) + D_{TV}(\tilde{p}_{\boldsymbol{\theta}}^r(\mathbf{x}_0|\mathbf{c})||q^r(\mathbf{x}_0|\mathbf{c})) \tag{21}$$

Now, we will investigate each term to determine the relationship between the lookahead error and the discretization steps $\delta$.

**First term $D_{TV}(p_{\boldsymbol{\theta}}^r(\mathbf{x}_0|\mathbf{c})||q^r(\mathbf{x}_0|\mathbf{c}))$.**

$$D_{TV}(p_{\boldsymbol{\theta}}^r(\mathbf{x}_0|\mathbf{c})||q^r(\mathbf{x}_0|\mathbf{c})) = \frac{1}{2}\int \left| \frac{p_{\boldsymbol{\theta}}(\mathbf{x}_0|\mathbf{c})w(\mathbf{x}_0, \mathbf{c})}{Z_p(\mathbf{c})} - \frac{q(\mathbf{x}_0|\mathbf{c})w(\mathbf{x}_0, \mathbf{c})}{Z_q(\mathbf{c})} \right| d\mathbf{x}_0 \tag{22}$$

$$= \frac{1}{2}\int \left| \frac{w(\mathbf{x}_0, \mathbf{c})\{p_{\boldsymbol{\theta}}(\mathbf{x}_0|\mathbf{c}) - q(\mathbf{x}_0|\mathbf{c})\}}{Z_p(\mathbf{c})} + q(\mathbf{x}_0|\mathbf{c})w(\mathbf{x}_0, \mathbf{c})\left\{ \frac{1}{Z_p(\mathbf{c})} - \frac{1}{Z_q(\mathbf{c})} \right\} \right| d\mathbf{x}_0 \tag{23}$$

$$\leq \frac{1}{2}\int \left| \frac{w(\mathbf{x}_0, \mathbf{c})\{p_{\boldsymbol{\theta}}(\mathbf{x}_0|\mathbf{c}) - q(\mathbf{x}_0|\mathbf{c})\}}{Z_p(\mathbf{c})} \right| d\mathbf{x}_0 + \frac{1}{2}\int \left| q(\mathbf{x}_0|\mathbf{c})w(\mathbf{x}_0, \mathbf{c})\left\{ \frac{1}{Z_p(\mathbf{c})} - \frac{1}{Z_q(\mathbf{c})} \right\} \right| d\mathbf{x}_0 \tag{24}$$

$$= \frac{1}{2Z_p(\mathbf{c})}\int w(\mathbf{x}_0, \mathbf{c})|p_{\boldsymbol{\theta}}(\mathbf{x}_0|\mathbf{c}) - q(\mathbf{x}_0|\mathbf{c})|\, d\mathbf{x}_0 + \frac{1}{2}\left| \frac{1}{Z_p(\mathbf{c})} - \frac{1}{Z_q(\mathbf{c})} \right| \int q(\mathbf{x}_0|\mathbf{c})w(\mathbf{x}_0, \mathbf{c})d\mathbf{x}_0 \tag{25}$$

$$= \frac{1}{2Z_p(\mathbf{c})}\int w(\mathbf{x}_0, \mathbf{c})|p_{\boldsymbol{\theta}}(\mathbf{x}_0|\mathbf{c}) - q(\mathbf{x}_0|\mathbf{c})|\, d\mathbf{x}_0 + \frac{1}{2Z_p(\mathbf{c})}|Z_p(\mathbf{c}) - Z_q(\mathbf{c})| \tag{26}$$

$$\leq \frac{1}{Z_p(\mathbf{c})}\int w(\mathbf{x}_0, \mathbf{c})|p_{\boldsymbol{\theta}}(\mathbf{x}_0|\mathbf{c}) - q(\mathbf{x}_0|\mathbf{c})|\, d\mathbf{x}_0 \tag{27}$$

$$\leq \frac{M}{m}\int |p_{\boldsymbol{\theta}}(\mathbf{x}_0|\mathbf{c}) - q(\mathbf{x}_0|\mathbf{c})|\, d\mathbf{x}_0 = \frac{2M}{m}D_{TV}(p_{\boldsymbol{\theta}}(\mathbf{x}_0|\mathbf{c})||q(\mathbf{x}_0|\mathbf{c})) \tag{28}$$

Eq. (24) is due to the triangle inequality and Eq. (27) is due to $|Z_p(\mathbf{c}) - Z_q(\mathbf{c})| = \left| \int w(\mathbf{x}_0, \mathbf{c})\{p_{\boldsymbol{\theta}}(\mathbf{x}_0|\mathbf{c}) - q(\mathbf{x}_0|\mathbf{c})\}d\mathbf{x}_0 \right| \leq \int w(\mathbf{x}_0, \mathbf{c})|p_{\boldsymbol{\theta}}(\mathbf{x}_0|\mathbf{c}) - q(\mathbf{x}_0|\mathbf{c})|\, d\mathbf{x}_0$. Eq. (28) holds due to $Z_p(\mathbf{c}) = \int p_{\boldsymbol{\theta}}(\mathbf{x}_0|\mathbf{c})w(\mathbf{x}_0, \mathbf{c})d\mathbf{x}_0 \geq \int m \cdot p_{\boldsymbol{\theta}}(\mathbf{x}_0|\mathbf{c})d\mathbf{x}_0 = m$ and $\int w(\mathbf{x}_0, \mathbf{c})|p_{\boldsymbol{\theta}}(\mathbf{x}_0|\mathbf{c}) - q(\mathbf{x}_0|\mathbf{c})|\, d\mathbf{x}_0 \leq M \cdot \int |p_{\boldsymbol{\theta}}(\mathbf{x}_0|\mathbf{c}) - q(\mathbf{x}_0|\mathbf{c})|\, d\mathbf{x}_0$. By the Theorem 4 in (Kim et al., 2022), we can conclude that:

$$D_{TV}(p_{\boldsymbol{\theta}}^r(\mathbf{x}_0|\mathbf{c})||q^r(\mathbf{x}_0|\mathbf{c})) \leq \mathcal{O}\left(1/\sqrt{\delta}\right) \tag{29}$$

**Second term $D_{TV}(\tilde{p}_{\boldsymbol{\theta}}^r(\mathbf{x}_0|\mathbf{c})||q^r(\mathbf{x}_0|\mathbf{c}))$.** By Pinsker's inequality, we have the upper bound of the second term:

$$D_{TV}(\tilde{p}_{\boldsymbol{\theta}}^r(\mathbf{x}_0|\mathbf{c})||q^r(\mathbf{x}_0|\mathbf{c})) \leq \sqrt{\frac{1}{2}D_{KL}(\tilde{p}_{\boldsymbol{\theta}}^r(\mathbf{x}_0|\mathbf{c})||q^r(\mathbf{x}_0|\mathbf{c}))} \tag{30}$$

By Theorem 2 in Song et al. (2021a), we have:

$$D_{KL}(\tilde{p}_{\boldsymbol{\theta}}^r(\mathbf{x}_0|\mathbf{c})||q^r(\mathbf{x}_0|\mathbf{c}))$$

$$= D_{KL}(\tilde{p}_{\boldsymbol{\theta}}^r(\mathbf{x}_T|\mathbf{c})||q^r(\mathbf{x}_T|\mathbf{c})) + \frac{1}{2}\int_0^T \mathbb{E}_{\tilde{p}_{\boldsymbol{\theta}}^r}(\mathbf{x}_t|\mathbf{c})\left[g_\sigma^2(t)\|\nabla_{\mathbf{x}_t}\log\tilde{p}_{\boldsymbol{\theta}}^r(\mathbf{x}_t|\mathbf{c}) - \nabla_{\mathbf{x}_t}\log q^r(\mathbf{x}_t|\mathbf{c})\|_2^2\right]dt \tag{31}$$

$$= D_{KL}(\tilde{p}_{\boldsymbol{\theta}}^r(\mathbf{x}_T|\mathbf{c})||q^r(\mathbf{x}_T|\mathbf{c})) + \frac{1}{2}\int_0^T \mathbb{E}_{\tilde{p}_{\boldsymbol{\theta}}^r}(\mathbf{x}_t|\mathbf{c})\left[g_\sigma^2(t)\|\nabla_{\mathbf{x}_t}\log p_{\boldsymbol{\theta}}(\mathbf{x}_t|\mathbf{c}) - \nabla_{\mathbf{x}_t}\log q(\mathbf{x}_t|\mathbf{c})\|_2^2\right]dt \tag{32}$$

$$= D_{KL}(\tilde{p}_{\boldsymbol{\theta}}^r(\mathbf{x}_T|\mathbf{c})||q^r(\mathbf{x}_T|\mathbf{c})) + \frac{1}{2}\int_0^T \int \frac{p_{\boldsymbol{\theta}}(\mathbf{x}_t|\mathbf{c})\mathbb{E}_{q(\mathbf{x}_0|\mathbf{x}_t)}[w(\mathbf{x}_0,\mathbf{c})]}{Z(\mathbf{c})}g^2(t)\|\nabla_{\mathbf{x}_t}\log p_{\boldsymbol{\theta}}(\mathbf{x}_t|\mathbf{c}) - \nabla_{\mathbf{x}_t}\log q(\mathbf{x}_t|\mathbf{c})\|_2^2 d\mathbf{x}_t dt \tag{33}$$

$$\leq D_{KL}(\tilde{p}_{\boldsymbol{\theta}}^r(\mathbf{x}_T|\mathbf{c})||q^r(\mathbf{x}_T|\mathbf{c})) + \frac{1}{2}\int_0^T \frac{M}{m}\int p_{\boldsymbol{\theta}}(\mathbf{x}_t|\mathbf{c})g_\sigma^2(t)\|\nabla_{\mathbf{x}_t}\log p_{\boldsymbol{\theta}}(\mathbf{x}_t|\mathbf{c}) - \nabla_{\mathbf{x}_t}\log q(\mathbf{x}_t|\mathbf{c})\|_2^2 d\mathbf{x}_t dt \tag{34}$$

$$= D_{KL}(\tilde{p}_{\boldsymbol{\theta}}^r(\mathbf{x}_T|\mathbf{c})||q^r(\mathbf{x}_T|\mathbf{c})) + \frac{M}{m}\mathcal{L}_{SM}(p_{\boldsymbol{\theta}}(\mathbf{x}_0|\mathbf{c}),q(\mathbf{x}_0|\mathbf{c});g_\sigma^2) \tag{35}$$

where $\mathcal{L}_{SM}(p_{\boldsymbol{\theta}}(\mathbf{x}_0|\mathbf{c}),q(\mathbf{x}_0|\mathbf{c});g_\sigma^2) := \frac{1}{2}\int_0^T \mathbb{E}_{p_{\boldsymbol{\theta}}(\mathbf{x}_t|\mathbf{c})}\left[g_\sigma^2(t)\|\nabla_{\mathbf{x}_t}\log p_{\boldsymbol{\theta}}(\mathbf{x}_t|\mathbf{c}) - \nabla_{\mathbf{x}_t}\log q(\mathbf{x}_t|\mathbf{c})\|_2^2\right]dt$ and $Z(\mathbf{c}) := \int p_{\boldsymbol{\theta}}(\mathbf{x}_t|\mathbf{c})\mathbb{E}_{q(\mathbf{x}_0|\mathbf{x}_t)}[w(\mathbf{x}_0,\mathbf{c})]d\mathbf{x}_0$. Eq. (34) holds due to $m \leq \mathbb{E}_{q(\mathbf{x}_0|\mathbf{x}_t)}[w(\mathbf{x}_0,\mathbf{c})] \leq M$ and $Z(\mathbf{c}) = \int p_{\boldsymbol{\theta}}(\mathbf{x}_t|\mathbf{c})\mathbb{E}_{q(\mathbf{x}_0|\mathbf{x}_t)}[w(\mathbf{x}_0,\mathbf{c})]d\mathbf{x}_0 \geq \int m \cdot p_{\boldsymbol{\theta}}(\mathbf{x}_t|\mathbf{c})d\mathbf{x}_0 = m$.

Then, we can get the upper bound of $\mathcal{L}_{SM}$ as follows:

$$\mathcal{L}_{SM}(p_{\boldsymbol{\theta}}(\mathbf{x}_0|\mathbf{c}),q(\mathbf{x}_0|\mathbf{c});g^2) = \frac{1}{2}\sum_{k=1}^\delta \int_{t_{k-1}}^{t_k} \mathbb{E}_{p_{\boldsymbol{\theta}}(\mathbf{x}_t|\mathbf{c})}\left[g_\sigma^2(t)\|\nabla_{\mathbf{x}_t}\log p_{\boldsymbol{\theta}}(\mathbf{x}_t|\mathbf{c}) - \nabla_{\mathbf{x}_t}\log q(\mathbf{x}_t|\mathbf{c})\|_2^2\right]dt \tag{36}$$

$$= \frac{1}{2}\sum_{k=1}^\delta \int_{t_{k-1}}^{t_k} \mathbb{E}_{p_{\boldsymbol{\theta}}(\mathbf{x}_t|\mathbf{c})}\left[g_\sigma^2(t)\left\|\nabla_{\mathbf{x}_t}\log p_{\boldsymbol{\theta}}(\mathbf{x}_t|\mathbf{c}) - \nabla_{\mathbf{x}_{t_k}}\log q(\mathbf{x}_{t_k}|\mathbf{c})\right\|_2^2\right]dt \tag{37}$$

$$= \frac{1}{2}\sum_{k=1}^\delta \int_{t_{k-1}}^{t_k} \mathbb{E}_{p_{\boldsymbol{\theta}}(\mathbf{x}_t|\mathbf{c})}\left[g_\sigma^2(t)\left\|\nabla_{\mathbf{x}_t}\log p_{\boldsymbol{\theta}}(\mathbf{x}_t|\mathbf{c}) - \nabla_{\mathbf{x}_{t_k}}\log p_{\boldsymbol{\theta}}(\mathbf{x}_{t_k}\mathbf{c})\right\|_2^2\right]dt \tag{38}$$

$$\leq \frac{1}{2}g_{max}^2 L^2 \sum_{k=1}^\delta \int_{t_{k-1}}^{t_k} \mathbb{E}_{p_{\boldsymbol{\theta}}(\mathbf{x}_t|\mathbf{c})}\|\mathbf{x}_t - \mathbf{x}_{t_k}\|_2^2 dt \tag{39}$$

$$\leq \frac{1}{2}g_{max}^2 L^2 \sum_{k=1}^\delta \int_{t_{k-1}}^{t_k} \sup_{u\in[0,T]} \mathbb{E}_{p_{\boldsymbol{\theta}}(\mathbf{x}_u|\mathbf{c})}\|\mathbf{x}_u - \hat{\mathbf{x}}_u\|_2^2 dt \tag{40}$$

$$\leq \frac{1}{2}g_{max}^2 L^2 \sum_{k=1}^\delta \int_{t_{k-1}}^{t_k} C\frac{T}{\delta} dt \tag{41}$$

$$= \frac{1}{2}g_{max}^2 L^2 C T^2 \frac{1}{\delta} \tag{42}$$

Eq. (38) is due to assumption 6, and Eq. (39) is due to assumptions 2 and 3. Eq. (41) holds due to the strong convergence of Euler-Maruyama method (Kloeden & Pearson, 1977; Higham, 2001; Ngo & Taguchi, 2016). Therefore, we have $D_{KL}(\tilde{p}_{\boldsymbol{\theta}}^r(\mathbf{x}_0|\mathbf{c})||q^r(\mathbf{x}_0|\mathbf{c})) \leq \mathcal{O}(1/\delta)$ and can conclude that:

$$D_{TV}(\tilde{p}_{\boldsymbol{\theta}}^r(\mathbf{x}_0|\mathbf{c})||q^r(\mathbf{x}_0|\mathbf{c})) \leq \mathcal{O}\left(1/\sqrt{\delta}\right) \tag{43}$$

In conclusion, combining Eq. (29) and Eq. (43) completes the proof. $\quad\square$

## A.5. Proof of Theorem 3.5

**Theorem 3.5.** *(Scaling Law for Lookahead Sample Size $n$) Under mild conditions, for all $\mathbf{x}_t$ and $\mathbf{c}$, the lookahead reward $\tilde{r}_t^\lambda$ and its empirical estimate $\hat{r}_t^\lambda$ satisfy:*

$$\tilde{r}_t^\lambda(\mathbf{x}_t, \mathbf{c}) - \hat{r}_t^\lambda(\mathbf{x}_t, \mathbf{c})$$
$$\xrightarrow{d} \mathcal{N}\left(0, \frac{\mathrm{Var}\left(p(\mathbf{x}_t \mid \mathbf{x}_0)\exp(\lambda \cdot r(\mathbf{x}_0, \mathbf{c}))\right)}{\sqrt{n}\left(\mathbb{E}_{q(\mathbf{x}_0|\mathbf{c})}\left[p(\mathbf{x}_t \mid \mathbf{x}_0)\right]\exp\left(\tilde{r}_t^\lambda(\mathbf{x}_t, \mathbf{c})\right)\right)^2}\right).$$

*Proof.* Recall the definitions:

$$\tilde{r}_t^\lambda(\mathbf{x}_t, \mathbf{c}) = \log \mathbb{E}_{q(\mathbf{x}_0|\mathbf{c})}\left[\frac{p(\mathbf{x}_t|\mathbf{x}_0)}{\mathbb{E}_{q(\mathbf{x}_0|\mathbf{c})}[p(\mathbf{x}_t|\mathbf{x}_0)]}\exp\left(\lambda \cdot r(\mathbf{x}_0, \mathbf{c})\right)\right], \quad \hat{r}_t^\lambda(\mathbf{x}_t, \mathbf{c}) := \log \frac{1}{n}\sum_{i=1}^n\left[\frac{p(\mathbf{x}_t|\hat{\mathbf{x}}_0^i)}{\frac{1}{n}\sum_{j=1}^n p(\mathbf{x}_t|\hat{\mathbf{x}}_0^j)}\exp\left(\lambda \cdot r(\hat{\mathbf{x}}_0^i, \mathbf{c})\right)\right].$$

We derive this result similarly to Theorem 1 in STF (Xu et al., 2023b) and make the following assumption:

1. Without loss of generality, suppose that $p(\mathbf{x}_t|\hat{\mathbf{x}}_0^j)$, $j = 2, ..., n$ are i.i.d. by conditioning with $\mathbf{x}_t \sim p(\mathbf{x}_t|\hat{\mathbf{x}}_0^1)$.

2. Suppose that $0 < \mathbb{E}_{q(\mathbf{x}_0|\mathbf{c})}[p(\mathbf{x}_t|\mathbf{x}_0)] < \infty$.

3. Suppose that $\mathrm{Var}(p(\mathbf{x}_t|\mathbf{x}_0)\exp\left(\lambda \cdot r(\mathbf{x}_0, \mathbf{c})\right)) < \infty$.

First, consider denominator term of $\exp(\hat{r}_t^\lambda(\mathbf{x}_t, \mathbf{c}))$. When $n \to \infty$, by the Weak Law of Large Numbers (WLLN),

$$\frac{1}{n}\sum_{j=1}^n p(\mathbf{x}_t \mid \hat{\mathbf{x}}_0^j) = \frac{p(\mathbf{x}_t \mid \hat{\mathbf{x}}_0^1)}{n} + \frac{n-1}{n}\cdot\frac{1}{n-1}\sum_{j=2}^n p(\mathbf{x}_t \mid \hat{\mathbf{x}}_0^j) \xrightarrow{p} \mathbb{E}_{q(\mathbf{x}_0|\mathbf{c})}[p(\mathbf{x}_t \mid \mathbf{x}_0)].$$

Next, consider the remaining term of $\exp(\hat{r}_t^\lambda(\mathbf{x}_t, \mathbf{c}))$. By Central Limit Theorem (CLT):

$$\sqrt{n}\left(\frac{1}{n}\sum_{i=1}^n p(\mathbf{x}_t \mid \hat{\mathbf{x}}_0^i)\exp(\lambda \cdot r(\hat{\mathbf{x}}_0^i, \mathbf{c})) - \mathbb{E}_{q(\mathbf{x}_0|\mathbf{c})}\left[p(\mathbf{x}_t \mid \mathbf{x}_0)\exp(\lambda \cdot r(\mathbf{x}_0, \mathbf{c}))\right]\right) \xrightarrow{d} \mathcal{N}\left(0, \mathrm{Var}\left(p(\mathbf{x}_t \mid \mathbf{x}_0)\exp(\lambda \cdot r(\mathbf{x}_0, \mathbf{c}))\right)\right).$$

Putting them together via Slutsky's theorem, we have:

$$\sqrt{n}\left(\exp(\hat{r}_t^\lambda(\mathbf{x}_t, \mathbf{c})) - \exp(\tilde{r}_t^\lambda(\mathbf{x}_t, \mathbf{c}))\right) \xrightarrow{d} \mathcal{N}\left(0, \frac{\mathrm{Var}\left(p(\mathbf{x}_t \mid \mathbf{x}_0)\exp(\lambda \cdot r(\mathbf{x}_0, \mathbf{c}))\right)}{\left\{\mathbb{E}_{q(\mathbf{x}_0|\mathbf{c})}[p(\mathbf{x}_t \mid \mathbf{x}_0)]\right\}^2}\right).$$

Finally, by the delta method with $g(x) := \log(x)$, we conclude the proof.

$$\tilde{r}_t^\lambda(\mathbf{x}_t, \mathbf{c}) - \hat{r}_t^\lambda(\mathbf{x}_t, \mathbf{c}) \xrightarrow{d} \mathcal{N}\left(0, \frac{\mathrm{Var}\left(p(\mathbf{x}_t \mid \mathbf{x}_0)\exp(\lambda \cdot r(\mathbf{x}_0, \mathbf{c}))\right)}{\sqrt{n}\left(\mathbb{E}_{q(\mathbf{x}_0|\mathbf{c})}\left[p(\mathbf{x}_t \mid \mathbf{x}_0)\right]\exp\left(\tilde{r}_t^\lambda(\mathbf{x}_t, \mathbf{c})\right)\right)^2}\right).$$

$\square$

## A.6. Analysis on Taylor Approximation Error

In gradient guidance methods (Bansal et al., 2024; Na et al., 2025), the reward function is approximated via a first-order Taylor expansion around $\bar{x}_0 = \mathbb{E}_{p_\theta(\mathbf{x}_0|\mathbf{x}_t,\mathbf{c})}[\mathbf{x}_0]$, which is obtained from Tweedie's formula.

$$r(\mathbf{x}_0, c) = r(\bar{\mathbf{x}}_0, c) + (\mathbf{x}_0 - \bar{\mathbf{x}}_0)^\top \nabla_{\mathbf{x}_0} r(\bar{\mathbf{x}}_0, c) + O(\|\mathbf{x}_0 - \bar{\mathbf{x}}_0\|^2) \tag{44}$$

The term $O(\|\mathbf{x}_0 - \bar{\mathbf{x}}_0\|^2)$ represents the higher-order error terms beyond the first-order expansion. Under this approximation, the expected future reward $r_t^\lambda(\mathbf{x}_t, \mathbf{c})$ defined in Eq. (5) can be expanded as follows:

$$r_t^\lambda(\mathbf{x}_t, \mathbf{c}) = \log \mathbb{E}_{p_\theta(\mathbf{x}_0|\mathbf{x}_t,c)}\big[\exp(\lambda \cdot r(\mathbf{x}_0, c))\big] \tag{45}$$

$$= \log \mathbb{E}_{p_\theta(\mathbf{x}_0|\mathbf{x}_t,c)}\Big[\exp\big(\lambda \cdot \big(r(\bar{\mathbf{x}}_0, c) + (\mathbf{x}_0 - \bar{\mathbf{x}}_0)^\top \nabla_{\mathbf{x}_0} r(\bar{\mathbf{x}}_0, c) + O(\|\mathbf{x}_0 - \bar{\mathbf{x}}_0\|^2)\big)\big)\Big] \tag{46}$$

$$= \log \mathbb{E}_{p_\theta(\mathbf{x}_0|\mathbf{x}_t,c)}\Big[\exp(\lambda \cdot r(\bar{\mathbf{x}}_0, c)) \exp\big(\lambda \cdot \big((\mathbf{x}_0 - \bar{\mathbf{x}}_0)^\top \nabla_{\mathbf{x}_0} r(\bar{\mathbf{x}}_0, c) + O(\|\mathbf{x}_0 - \bar{\mathbf{x}}_0\|^2)\big)\big)\Big] \tag{47}$$

$$= \log \Big( \exp(\lambda \cdot r(\bar{\mathbf{x}}_0, c)) \mathbb{E}_{p_\theta(\mathbf{x}_0|\mathbf{x}_t,c)}\big[\exp\big(\lambda \cdot \big((\mathbf{x}_0 - \bar{\mathbf{x}}_0)^\top \nabla_{\mathbf{x}_0} r(\bar{\mathbf{x}}_0, c) + O(\|\mathbf{x}_0 - \bar{\mathbf{x}}_0\|^2)\big)\big)\big]\Big) \tag{48}$$

$$= \underbrace{\lambda \cdot r(\bar{\mathbf{x}}_0, c)}_{\text{Taylor approximation}} + \underbrace{\log \mathbb{E}_{p_\theta(\mathbf{x}_0|\mathbf{x}_t,c)}\big[\exp\big(\lambda \cdot \big((\mathbf{x}_0 - \bar{\mathbf{x}}_0)^\top \nabla_{\mathbf{x}_0} r(\bar{\mathbf{x}}_0, c) + O(\|\mathbf{x}_0 - \bar{\mathbf{x}}_0\|^2)\big)\big)\big]}_{\text{Taylor approximation error } (:=\epsilon_{\text{Taylor}})}. \tag{49}$$

We denote the error induced by the Taylor approximation as $\epsilon_{\text{Taylor}}$. Then, by Jensen's inequality,

$$\epsilon_{\text{Taylor}} \geq \mathbb{E}_{p_\theta(\mathbf{x}_0|\mathbf{x}_t,c)}\Big[\lambda \cdot \big((\mathbf{x}_0 - \bar{\mathbf{x}}_0)^\top \nabla_{\mathbf{x}_0} r(\bar{\mathbf{x}}_0, c) + O(\|\mathbf{x}_0 - \bar{\mathbf{x}}_0\|^2)\big)\Big] \tag{50}$$

$$= \lambda \cdot \underbrace{\mathbb{E}_{p_\theta(\mathbf{x}_0|\mathbf{x}_t,c)}\Big[(\mathbf{x}_0 - \bar{\mathbf{x}}_0)^\top \nabla_{\mathbf{x}_0} r(\bar{\mathbf{x}}_0, c) + O(\|\mathbf{x}_0 - \bar{\mathbf{x}}_0\|^2)\Big]}_{\text{Constant with respect to } \lambda} \tag{51}$$

$$= \lambda \cdot C. \tag{52}$$

When the reward function is approximated via a Taylor expansion, the resulting error in the expected future reward $\epsilon_{\text{Taylor}}$ is lower bounded by $\lambda \cdot C$ for some constant $C$, where $C$ is a constant independent of $\lambda$. Consequently, as $\lambda$ increases, i.e., as the guidance strength becomes stronger, this approximation error inevitably grows. For this reason, gradient guidance methods typically cannot employ a large guidance strength without incurring significant approximation errors.

### A.7. Extension to Discrete Diffusion Models

Consider $\mathbf{x}_0 := [x_0^1, \ldots, x_0^L]$ as a discrete sequence to be generated. The denoising formulation corresponding to Eq. (5) is given by:

$$p_\theta^r(\mathbf{x}_t|\mathbf{c}, \mathbf{x}_{t+1}) \propto p_\theta(\mathbf{x}_t|\mathbf{c}, \mathbf{x}_{t+1}) \cdot \exp\big(r_t^\lambda(\mathbf{x}_t, \mathbf{c}) - r_{t+1}^\lambda(\mathbf{x}_{t+1}, \mathbf{c})\big)$$
$$\propto p_\theta(\mathbf{x}_t|\mathbf{c}, \mathbf{x}_{t+1}) \cdot \exp\big(r_t^\lambda(\mathbf{x}_t, \mathbf{c})\big)$$
$$\approx p_\theta(\mathbf{x}_t|\mathbf{c}, \mathbf{x}_{t+1}) \cdot \exp\big(\hat{r}_t^\lambda(\mathbf{x}_t, \mathbf{c}).\big)$$

The discrete diffusion model generally generates each token independently, and following Discrete Classifier Guidance (Schiff et al., 2025), we also compute the empirical reward term in a token-independent manner.

$$p_\theta^r(x_t^{(l)}|\mathbf{c}, \mathbf{x}_{t+1}) \propto \underbrace{p_\theta(x_t^{(l)}|\mathbf{c}, \mathbf{x}_{t+1})}_{\text{Base Diffusion}} \cdot \exp\Big(\hat{r}_t^\lambda(x_t^{(l)}, \mathbf{c}).\Big)$$

We derive the token-independent reward term as follows:

$$\hat{r}_t^\lambda\big(x_t^{(l)}, \mathbf{c}\big) := \log \frac{1}{n} \sum_{i=1}^{n}\left[ \frac{p\big(x_t^{(l)} \mid \hat{\mathbf{x}}_0^{(1:L),i}\big)}{\frac{1}{n}\sum_{j=1}^{n} p\big(x_t^{(l)} \mid \hat{\mathbf{x}}_0^{(1:L),j}\big)} \exp\Big(\lambda \cdot r\big(\hat{\mathbf{x}}_0^{(1:L),i}, \mathbf{c}\big)\Big)\right] \tag{53}$$

$$= \log \frac{1}{n} \sum_{i=1}^{n}\left[ \frac{p\big(x_t^{(l)} \mid \hat{\mathbf{x}}_0^{(l),i}\big)}{\frac{1}{n}\sum_{j=1}^{n} p\big(x_t^{(l)} \mid \hat{\mathbf{x}}_0^{(l),j}\big)} \exp\Big(\lambda \cdot r\big(\hat{\mathbf{x}}_0^{(1:L),i}, \mathbf{c}\big)\Big)\right]. \tag{54}$$

Although rewards are assigned at the sequence level, the resulting denoising process remains token-independent. We believe this may limit the ability of the guidance term to capture sequence-level information. Developing guidance mechanisms that more effectively incorporate sequence-level information into the denoising process is a promising direction for future work.

# B. Experimental Setting

## B.1. Additional Details on LiDAR Configuration

We use $\lambda = 5000$ in all experiments unless otherwise specified, as the left panel of Figures 8 and 13 shows that sufficiently large values of $\lambda$ improve all metrics. We set $s = 12.5$ for SD v1.5, $s = 8$ for SDXL, and $s = 40$ for FLUX by default. The left two panels of Figure 3 report performance averaged over $s \in \{12.5, 15, 17.5\}$, while Figure 4a ablates $s \in \{0.5, 1.0, 2.5, 5.0, 7.5, 12.5, 15.0, 17.5\}$. As shown in Figure 4a, increasing $s$ to achieve higher IR also leads to higher validation metrics, including CLIP and HPS. Based on these observations, we select $s$ to maximize IR for evaluation on GenEval. For SDv1.5 and SDXL, we apply LiDAR only during the early stage of the denoising process, corresponding to the interval $[1.0, 0.2]$. For FLUX, LiDAR is applied only in $[1.0, 0.95]$. We use $\lambda = 1.5$ and $s = 1$ for UDLM experiments. We use pre-trained UDLM (Schiff et al., 2025) diffusion model.

## B.2. Baseline Configuration

**Universal guidance (UG) (Bansal et al., 2024):**

We use the Forward Universal Guidance approach, which applies gradient guidance to $\mathbf{x}_t$. Figure 4a shows ablation results on SD v1.5 for UG with respect to $\lambda$ over the range $\{0.5, 1.0, 2.5, 5.0, 7.5, 12.5, 15.0, 17.5\}$. The results indicate that increasing IR leads to decreased validation metrics, such as HPS. Although the validation performance is maximized at $\lambda = 0$ in UG, this choice reduces to vanilla sampling. Therefore, we use the value of $\lambda$ that yields the highest IR as the default for UG. Accordingly, we set $\lambda = 15$ as the default for SD v1.5 (and apply this value to the DDIM experiments). For SDXL, we evaluate $\lambda \in \{1, 2, 3, 4, 6\}$ with batch size 1 and select $\lambda = 3$, which yields the highest IR; thus, we fix $\lambda = 3$.

**DATE (Na et al., 2025):**

DATE partially mitigates the reward hacking observed in UG by updating the text embedding instead of $\mathbf{x}_t$, thereby providing a setting in which all validation metrics improve. For SD v1.5, we sweep $\rho$ (the counterpart of $\lambda$ in UG for the text embedding space; see Eq. 7 of DATE (Na et al., 2025) for the definition) over $\{0.05, 0.1, 0.5\}$ and select the value that yields the highest IR without decreasing CLIP or HPS. DATE achieves the highest IR at $\rho = 0.5$ under the maximum scaling setting; however, this choice also reduces HPS compared to vanilla sampling. In contrast, $\rho = 0.1$ yields balanced improvements in IR and other validation metrics, and we therefore select this value for evaluation on GenEval. Similarly, for SDXL, we sweep $\rho$ over $\{0.5, 1, 2, 3, 6, 10\}$ with batch size 1 and select $\rho = 2$.

Applying gradient updates at every time step incurs a higher computational cost than UG (58.80 sec vs. 58.36 sec on SD v1.5, and 498.96 sec vs. 334.43 sec on SDXL), since DATE requires one additional conditional Stein score evaluation with an updated text embedding at each step (Regarding memory, the backward pass requires slightly less memory, since it does not include the unconditional Stein score in CFG). However, DATE allows updates to be applied intermittently rather than at every step, enabling more efficient variants. To clearly present the efficiency–performance trade-off improvements from UG to DATE, and from DATE to LiDAR, we use DATE with text embedding updates applied every other step as the default baseline in Table 2, and evaluate its scaling behavior with respect to update frequency of $\{1, 2, 3, 5, 10\}$ in Figures 4b and 4c.

**SMC (Singhal et al., 2025):**

We follow the SMC implementation from FK steering (Singhal et al., 2025), which performs importance resampling every 20 denoising steps. To remain faithful to the formulation, we exclude additional heuristics such as adaptive resampling in the experiments of Table 8. For the diversity comparison in Figures 11 and 12, adaptive resampling slightly increases diversity from 0.090 to 0.156; however, the results remain substantially more collapsed compared to LiDAR.

**Sample-based diffusion guidance (Kim et al., 2025a; Kirchhof et al., 2025):**

Our method performs guidance during denoising by leveraging lookahead samples. Several methods with a sample-based guidance, such as Safe-D (Kim et al., 2025a) and SR (Kirchhof et al., 2025), have been introduced in the main paper. Both approaches rely on a reference set $\{\mathbf{x}_0^1, ... \mathbf{x}_0^K\}$ and construct guidance term $\Delta$ on $\bar{\mathbf{x}}_0 := \mathbb{E}_{\mathbf{x}_0 \sim p_\theta(\mathbf{x}_0 | \mathbf{x}_t, c)}[\mathbf{x}_0]$ that pushes samples away from this set. This guidance is incorporated by updating the estimate as $\bar{\mathbf{x}}_0 + \Delta$.

In our setting, we construct the reference set using lookahead samples with low reward values, and examine whether applying these guidance mechanisms can also lead to high-reward sampling.

Safe-D (Kim et al., 2025a) aims to construct a safe denoiser by leveraging unsafe samples. In the $\mathbf{x}_0$-space, it applies the

following guidance term:

$$\Delta = s \cdot \eta \, \beta(\mathbf{x}_t) \Big( \bar{\mathbf{x}}_0 - \sum_{i=1}^{K} \mathbf{x}_0^i \frac{p(\mathbf{x}_t \mid \mathbf{x}_0^i)}{\sum_{j=1}^{K} p(\mathbf{x}_t \mid \mathbf{x}_0^j)} \Big), \tag{55}$$

where $\beta(\mathbf{x}_t) = \frac{1}{K} \sum_{i=1}^{K} p(\mathbf{x}_t \mid \mathbf{x}_0^i)$. The parameters $s$ and $\eta$ control the overall guidance scale, and $K$ denotes the number of reference samples. Although many text-to-image models perform such manipulations in the latent space, the extremely high dimensionality often causes $\beta(\mathbf{x}_t)$ to become nearly zero. Since the paper does not provide official code, the exact implementation details were unclear. Therefore, we treated the scaling term $\beta := \eta \, \beta(\mathbf{x}_t)$ as a tunable hyperparameter and adjusted its value in our experiments. We conducted a hyperparameter search over combinations of $(s, K, \beta)$ where

$$s \in \{1, 10\}, \ K \in \{15, 25\}, \ \beta \in \{0.05, 0.1, 0.15, 0.2, 0.25\} \tag{56}$$

The best-performing setting reported in Table 7 is $(s, K, \beta) = (1, 15, 0.05)$.

SR (Kirchhof et al., 2025) aims to guide samples away from a given reference set. In the $\mathbf{x}_0$-space, it applies the following guidance term:

$$\Delta = s \cdot \sum_{i=1}^{K} \mathrm{ReLU}\left( \frac{r}{\|\bar{\mathbf{x}}_0 - \mathbf{x}_0^i\|} - 1 \right) (\bar{\mathbf{x}}_0 - \mathbf{x}_0^i). \tag{57}$$

Here, the parameter $s$ controls the guidance scale. $K$ denotes the number of reference samples, and $r$ is a radius that encourage the generated sample to move further away from the ball with radius $r$ centered on reference sample.

We conducted a hyperparameter search over combinations of $(s, K, r)$, where

$$s \in \{1, 10, 17.5\}, \ K \in \{5, 10, 15, 20, 25, 30, 35, 40, 45, 50\}, \ r \in \{50, 75, 100\}. \tag{58}$$

The best-performing setting reported in Table 7 is $(s, K, r) = (1, 40, 100)$.

Both experiments indicate that simply guiding samples away from low-reward regions has inherent limitations in achieving high-reward generation. To further support this claim, we conduct a hyperparameter analysis. Figure 9 illustrates the hyperparameter trends of SR (Kirchhof et al., 2025). Starting from the best-performing configuration $(s, K, r) = (1, 40, 100)$, we vary one hyperparameter at a time:

$$s \in \{0.2, 0.4, 0.6, 0.8, 1, 3, 5\}, \ K \in \{5, 10, 15, 20, 25, 30, 35, 40\}, \ r \in \{80, 90, 100, 110, 120\}. \tag{59}$$

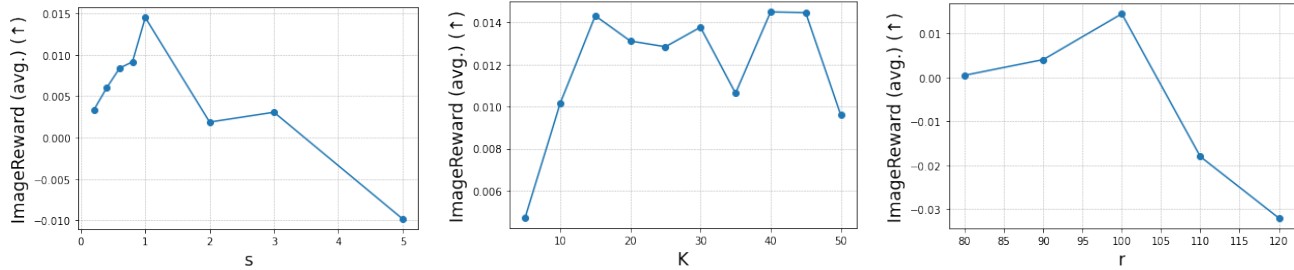

*(a)* The effect of varying $s$ on ImageReward.  *(b)* The effect of varying $K$ on ImageReward.  *(c)* The effect of varying $r$ on ImageReward.

*Figure 9.* Analysis of ImageReward trends by varying a single parameter of SR (Kirchhof et al., 2025) while others are fixed.

For the guidance strength parameter $s$, Increasing $s$ initially improves the reward, but overly strong guidance pushes samples outside the data manifold, resulting in degraded performance as shown in Figure 9a.

Figure 9b shows that using too small number of reference samples $K$ it cannot helps the model escape low-reward regions.

When varying the radius $r$, increasing $r$ allows the model to escape low-reward regions more effectively, consistent with Figure 9c. Nevertheless, when $r$ becomes excessively large, the generated samples may drift away from the data manifold, leading to a decrease in reward.

Overall, these observations demonstrate that guidance strategies that solely encourage moving away from low-reward samples are not sufficient to achieve high-reward sampling.

### B.3. Evaluation Metrics

We measure the following metrics across all experiments on 553 GenEval prompts (Ghosh et al., 2023).

**ImageReward** (Xu et al., 2023a): To assess the overall quality from a human-centric perspective, we utilize ImageReward. Unlike standard metrics, this learned model is trained on large-scale human preference data to output a scalar score representing the generated image's desirability. It effectively balances prompt relevance with visual aesthetics, capturing the general human consensus on image quality rather than focusing on a single dimension of fidelity.

**CLIP Score** (Radford et al., 2021): We employ CLIP Score to measure the high-level semantic correspondence between the text prompt and the generated image. By calculating the cosine similarity between their respective embeddings, this metric evaluates global semantic alignment without requiring reference images. However, it is generally less effective at distinguishing fine-grained details, such as precise object counts, spatial arrangements, or complex attribute binding.

**HPS (Human Preference Score)** (Wu et al., 2023a): HPS is adopted to evaluate the generation quality based on learned human preferences. By training a vision–language backbone on pairwise comparisons, this metric predicts which image humans are more likely to favor. It prioritizes *subjective preference* and *overall perceptual quality*, reflecting how aesthetic appeal and prompt satisfaction interact, rather than purely measuring semantic similarity at the embedding level.

**GenEval** (Ghosh et al., 2023): For a rigorous assessment of structural accuracy, we use GenEval. This object-centric protocol employs detection models to verify whether the entities and attributes specified in the prompt are correctly generated. By focusing on *compositional correctness*—such as object presence, counts, and spatial relations—GenEval serves as a crucial diagnostic tool for complex instruction-following failures that coarse semantic metrics often overlook.

### B.4. How to Calculate EFR Approximation Error in Figure 8

We define the *Expected Future Reward* (EFR) in Eq. 5 as

$$r_t^\lambda(\mathbf{x}_t, \mathbf{c}) := \log \mathbb{E}_{p_\theta(\mathbf{x}_0|\mathbf{x}_t, \mathbf{c})} \left[ \exp\left(\lambda \cdot r(\mathbf{x}_0, \mathbf{c})\right) \right]. \tag{60}$$

Using Theorem 3.1 and Monte-Carlo approximation, this quantity can be reformulated as

$$r_t^\lambda(\mathbf{x}_t, \mathbf{c}) = \log \mathbb{E}_{p_\theta(\mathbf{x}_0|\mathbf{c})} \left[ \frac{p(\mathbf{x}_t \mid \mathbf{x}_0)}{\mathbb{E}_{p_\theta(\mathbf{x}_0|\mathbf{c})}[p(\mathbf{x}_t \mid \mathbf{x}_0)]} \exp\left(\lambda \cdot r(\mathbf{x}_0, \mathbf{c})\right) \right] \tag{61}$$

$$\approx \log \frac{1}{N} \sum_{i=1}^{N} \left[ \frac{p(\mathbf{x}_t \mid \mathbf{x}_0^i)}{\frac{1}{N} \sum_{j=1}^{N} p(\mathbf{x}_t \mid \mathbf{x}_0^j)} \exp\left(\lambda \cdot r(\mathbf{x}_0^i, \mathbf{c})\right) \right], \tag{62}$$

where $\mathbf{x}_0^i \sim p_\theta(\mathbf{x}_0 \mid \mathbf{c})$. We refer to this Monte-Carlo estimate as $\text{EFR}_{\text{True}}(\mathbf{x}_t, t)$, which is used for evaluation. For this approximation, we generate $N = 100$ samples using a DDPM solver with 100 denoising steps, consistent with Table 2.

Gradient-guidance approaches (Bansal et al., 2024) approximate the EFR via a first-order Taylor expansion around the posterior mean, yielding

$$r_t^\lambda(\mathbf{x}_t, \mathbf{c}) \approx r(\bar{\mathbf{x}}_0, c), \tag{63}$$

where $\bar{\mathbf{x}}_0 := \mathbb{E}_{\mathbf{x}_0 \sim p_\theta(\mathbf{x}_0|\mathbf{x}_t, c)}[\mathbf{x}_0]$. We denote this approximation as $\text{EFR}_{\text{Grad}}(\mathbf{x}_t, t)$.

LiDAR approximates the EFR using the same Monte-Carlo form in Eq. 62, but draws samples from $\mathbf{x}_0^i \sim q(\mathbf{x}_0 \mid \mathbf{c})$, where $q$ denotes the distribution induced by a fast generator that produces lookahead samples. In this approximation, we generate $N = 800$ lookahead samples using DPM-5 solver. We denote this approximation as $\text{EFR}_{\text{LiDAR}}(\mathbf{x}_t, t)$.

We generate $K$ sample trajectories along diffusion timesteps $T$ with LiDAR guidance and evaluate $\text{EFR}_{\text{Grad}}(\mathbf{x}_t^k, t)$ and $\text{EFR}_{\text{LiDAR}}(\mathbf{x}_t^k, t)$ along each trajectory. We then compute the accumulated approximation error of $\text{EFR}_{\text{Approx}}(\mathbf{x}_t^k, t) \in \{\text{EFR}_{\text{Grad}}(\mathbf{x}_t^k, t), \text{EFR}_{\text{LiDAR}}(\mathbf{x}_t^k, t)\}$ with Mean Squared Error (MSE):

$$\frac{1}{K} \sum_{k=1}^{K} \sum_{t=0}^{T} \left( \text{EFR}_{\text{True}}(\mathbf{x}_t^k, t) - \text{EFR}_{\text{Approx}}(\mathbf{x}_t^k, t) \right)^2 \tag{64}$$

In this experiment, we use $T = 100$, and $K = 20$. In Figure 8, we examine how the accumulated approximation error changes as we vary $\lambda$.

As discussed in Appendix A.6, the error introduced by the Taylor approximation grows with $\lambda$. Therefore, under gradient guidance, the EFR approximation error consistently increases as $\lambda$ becomes larger. In contrast, our method does not suffer from Taylor approximation error and only incurs lookahead sampling error. As a result, our EFR approximation remains robust even when $\lambda$ increases.

# C. Related Work

**Formulation of expected future reward:** Reward-tilted sampling has been studied from multiple perspectives. The reward-tilted distribution arises as the optimal policy of KL-regularized reinforcement learning, which underlies RLHF (Ouyang et al., 2022) and related approaches such as DPO (Rafailov et al., 2023; Kim et al., 2025b). In diffusion models, realizing this target distribution requires characterizing the marginal distribution at each timestep. ELEGANT (Uehara et al., 2024) derives the intermediate marginal distributions that lead to the reward-tilted target distribution at $t = 0$, while AM (Domingo-Enrich et al., 2025) provides a stochastic optimal control interpretation of the resulting formulation. FK-Steering (Singhal et al., 2025) further derives a general discrete-time formulation of the corresponding marginal distributions.

**How to compute expected future reward:** This line of work focuses on efficiently computing the expected future reward. *Backward rollout* is generally infeasible due to its high computational cost. GlassFlow (Holderrieth et al., 2026) partially alleviates this issue by replacing stochastic sampling with ODE sampling, while MFM (Potaptchik et al., 2026) further improves efficiency by replacing the rollout sampler with a distilled model. TILT (Potaptchik et al., 2025) additionally proposes a backpropagation-free guidance method within the backward rollout framework. In contrast, LiDAR reformulates expected future reward computation through *forward rollout* rather than backward rollout. This reformulation naturally enables ODE-based lookahead sampling, distilled lookahead sampling, and backpropagation-free guidance within a unified framework. Another line of work relies on Tweedie's formula (Chung et al., 2023) to improve computational efficiency, but such approaches introduce approximation bias. Moreover, because no derivative-free guidance method is known under the Tweedie-based formulation, additional sampling strategies such as gradient guidance (Bansal et al., 2024; Na et al., 2025) and SMC (Singhal et al., 2025; Li et al., 2025) have been developed, each introducing its own limitations.

# D. Additional Experimental Results

## D.1. Efficiency Analysis

Table 9 presents a component-wise breakdown of the computational cost of LiDAR. Note that the cost of target sampling in Algorithm 2 is largely identical to that of vanilla sampling, as illustrated in Figure 10. Using up to $n < 800$ lookahead samples incurs nearly identical memory and runtime costs, consistent with observations in Safe-D (Kim et al., 2025a).

*Table 9.* Computational cost for each component for LiDAR (DPM-5 / $n$=50) in Table 2.

| Component | Time (sec.) |
|---|---|
| Lookahead sampling | 5.69 |
| Reward annotation | 0.65 |
| Target sampling | 7.07 |
| Total | 13.41 |

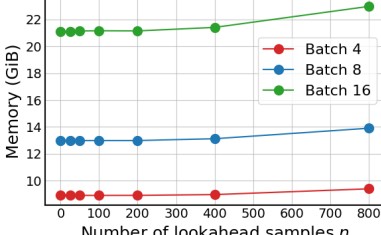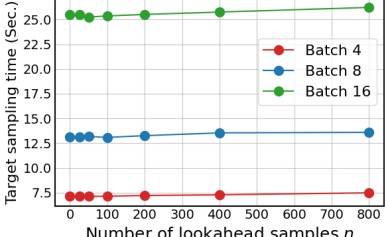

*Figure 10.* The inference cost for target sampling (Algorithm 2) according to the number of lookahead samples $n$ for SD v1.5.

## D.2. Robust Scaling Behavior with respect to $\delta$

Figure 11 shows that LiDAR exhibits robust scaling behavior with respect to the number of steps $\delta$ in the lookahead solver. Smaller values of $\delta$ incur larger lookahead error but permit a greater number of lookahead samples $n$ under the same computational budget, and vice versa. This trade-off results in nearly identical scaling behavior across different choices of $\delta$. We observe that the DPM-5 lookahead solver performs slightly better on SD v1.5; therefore, we adopt it as the default choice.

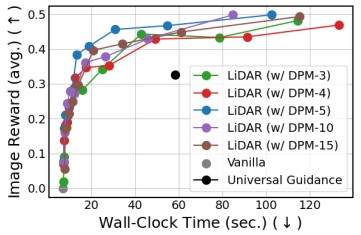
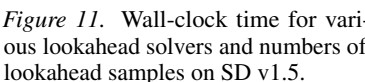

*Figure 11.* Wall-clock time for various lookahead solvers and numbers of lookahead samples on SD v1.5.

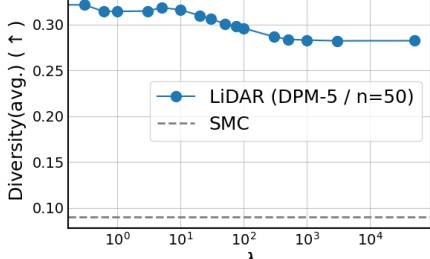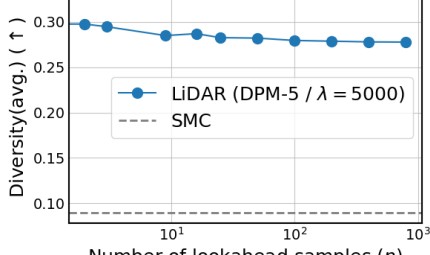

*Figure 12.* Diversity as a function of $\lambda$ and $n$ for SD v1.5.

## D.3. Diversity Analysis

Larger values of $\lambda$ consistently improve reward alignment performance, as shown in Figure 8, but slightly reduce diversity, as illustrated in Figure 12. Diversity is measured using pairwise similarity in the CLIP embedding space, following the protocol of FK. [2] The left panel of Figure 12 indicates that this reduction in diversity is minimal compared to the issues observed in SMC (Singhal et al., 2025). Moreover, all LiDAR-generated results in Figure 2 use the same value of $\lambda = 5000$ and still maintain substantial diversity.

Another factor influencing diversity is the number of lookahead samples $n$. Larger values of $n$ similarly improve reward alignment, as shown in Figure 3, but also lead to reduced diversity, as illustrated in the right panel of Figure 12. The diversity difference between LiDAR (DPM-3, $n = 3$) and LiDAR (DPM-5, $n = 50$) in Figure 2 primarily arises from this difference in $n$. This effect may stem from the need to avoid a larger number of low-reward lookahead samples, which restricts the feasible sampling region. Nevertheless, LiDAR continues to generate diverse samples, in contrast to the limitations observed in SMC. Finally, we note that fidelity and diversity are fundamentally traded off in generative modeling, and LiDAR provides effective control through the choice of $\lambda$ and $n$.

---

[2] https://github.com/zacharyhorvitz/Fk-Diffusion-Steering/blob/main/text_to_image/fkd_diffusers/rewards.py

## D.4. SDXL with DDIM 50 Steps

Table 10 shows results using the SDXL backbone with a DDIM-50 target sampler. We observe that UG and DATE exhibit more pronounced performance degradation, whereas LiDAR experiences a smaller performance drop compared to the DDPM-100 step sampler.

*Table 10.* All information is provided in Table 2

| Backbone | Guidance Method | Performance (↑) | | | | Inference Cost (↓) | |
|---|---|---|---|---|---|---|---|
| | | ImageReward | CLIP Score | HPS | GenEval | Time (sec.) | Mem. (GiB) |
| | Vanilla | 0.538 | 0.279 | 0.278 | 0.510 | 21.45 | 33.84 |
| | Powerful Solver (200 step) | 0.586 | 0.279 | 0.283 | 0.517 | 83.14 | 33.84 |
| | Universal Guidance | 0.643 | 0.277 | 0.277 | 0.512 | 169.33 | OOM* |
| | DATE | 0.722 | 0.280 | 0.281 | 0.539 | 138.47 | OOM* |
| SDXL w/ DDIM (50 step solver) | LiDAR (DPM-8 / $n$=3) | 0.749 | 0.282 | 0.287 | 0.530 | 24.78 | **33.84** |
| | LiDAR (DPM-8 / $n$=9) | 0.824 | 0.282 | 0.291 | 0.551 | 31.49 | **33.84** |
| | LiDAR (DPM-8 / $n$=50) | 0.950 | 0.284 | 0.293 | 0.585 | 77.44 | **33.84** |
| | LiDAR (LCM-4 / $n$=3) | 0.729 | 0.281 | 0.288 | 0.544 | 23.03 | **33.84** |
| | LiDAR (LCM-4 / $n$=16) | 0.820 | 0.283 | 0.291 | 0.574 | 29.60 | **33.84** |
| | LiDAR (LCM-4 / $n$=100) | 0.935 | **0.285** | 0.294 | **0.590** | 72.63 | **33.84** |
| | LiDAR (DMD-1 / $n$=3) | 0.706 | 0.281 | 0.287 | 0.539 | **22.57** | **33.84** |
| | LiDAR (DMD-1 / $n$=16) | 0.850 | 0.283 | 0.294 | 0.576 | 27.27 | **33.84** |
| | LiDAR (DMD-1 / $n$=100) | **0.954** | 0.284 | **0.297** | 0.588 | 58.12 | **33.84** |

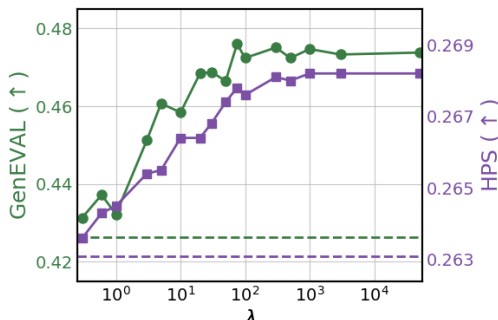

*Figure 13.* $\lambda$ ablations on HPS and GenEval.

## D.5. Sample Comparison

Figure 14 shows additional samples generated for other prompts.

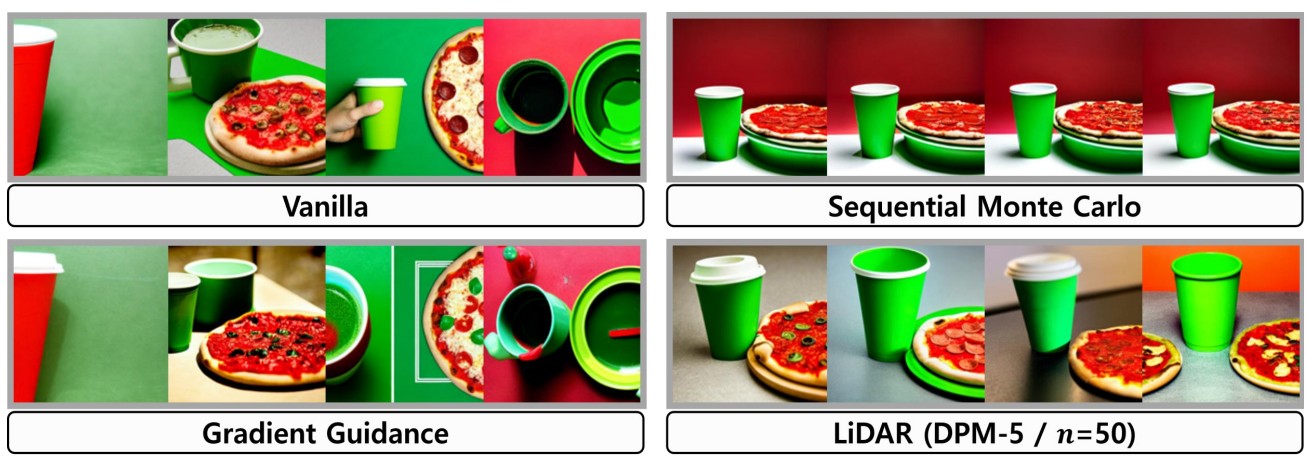

**Prompt:** *a photo of a green cup and a red pizza*

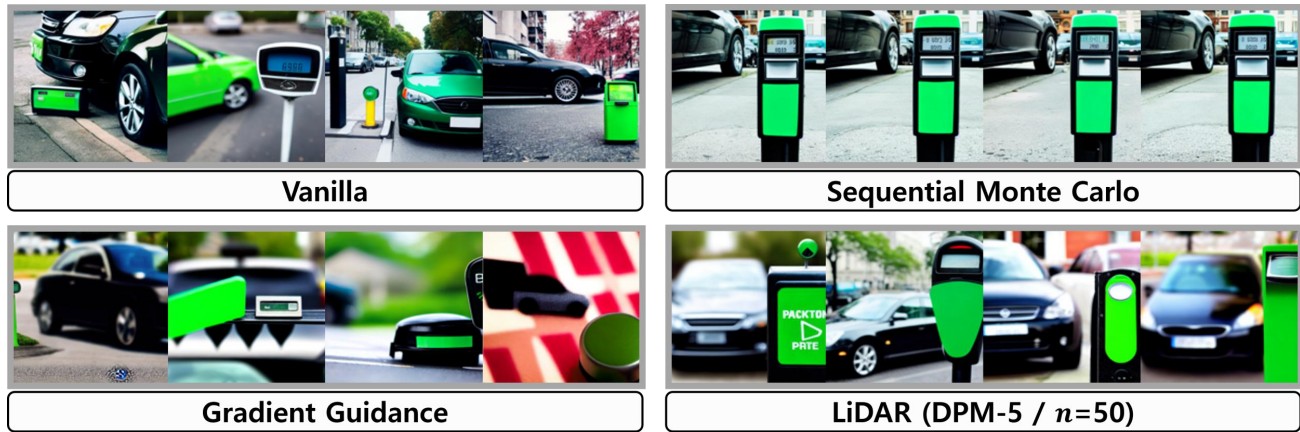

**Prompt:** *a photo of a black car and a green parking meter*

*Figure 14.* Visual comparison of generated samples with different prompts using SD v1.5.

