# OpenReview forum: "Lookahead Sample Reward Guidance for Test-Time Scaling of Diffusion Models"
_ICML.cc/2026/Conference — ICML 2026 spotlight_

### Official Review · Reviewer_MU56 · 2026-03-04

**Soundness:** 3
**Presentation:** 3
**Significance:** 3
**Originality:** 3
**Overall Recommendation:** 5
**Confidence:** 4

**Summary:**

This paper addresses scalable test-time reward alignment for diffusion models without additional training or backpropagation through the diffusion model. The proposed approach first generates a batch of look-ahead samples from a pretrained model and then applies closed-form, neural-gradient-free guidance at each diffusion step, yielding substantial inference-time efficiency gains over gradient-based guidance baselines. Experiments on modern text-to-image diffusion models demonstrate improved reward alignment compared to standard diffusion sampling and gradient-guidance methods.

**Compliance With Llm Reviewing Policy:**

Affirmed.

**Final Justification:**

The authors addressed all concerns and questions raised in the initial review. The additional results and the authors' clarifications have strengthened the submission. The strengths of this work now outweigh its weaknesses and limitations. All factors considered, I have raised my score and recommend a clear accept.

**Key Questions For Authors:**

## Questions and Other Comments

### Questions

1. A direct comparison with SMC methods seems to be missing. The authors discuss SMC methods by integrating them with LiDAR and argue that SMC is not comparable to LiDAR under the current experimental setup. However, $N$ appears to play a similar and central role in SMC as n does in LiDAR. One potentially "fair" setup for direct comparison would be to set $N=n$ (subject to memory constraints) and report the average performance across all $N$ generated samples. For Best-of-N evaluation, it would also be informative to set $N=n$ and evaluate the best $K$ out of $N$, with $K=4$ to match the current setting.
2. The alignment target appears to be strongly (positively) correlated with the purpose of classifier-free guidance (CFG): both IR and CLIP primarily measure text–image semantic alignment. Under the current setup, LiDAR may look like only facilitating text prompt conditioning. However, the proposed method should plausibly apply more broadly (including to unconditional models), where the alignment target differs from (though does not conflict with) the conditioning objective $c$. For example, one could use a neutral conditioning prompt such as "a photo of one human face", while defining a reward that assigns higher scores to images containing humans with particular gender(s) or age groups (e.g., ≥60). Could the authors include results under such a setting, where the condition and reward are more weakly correlated? If LiDAR performs well, this would substantially strengthen the paper’s claims; if not, the authors should discuss possible reasons and clearly state the limitation.
3. The value of $n$ appears to be critical for performance (given a fixed look-ahead sampling scheme), but it is not adequately discussed in the experiments as only three values are reported in Table 3. Could the authors provide additional tables or figures showing the relationship between $n$ and key metrics, analogous to Table 6 and the rightmost plot in Figure 7, but across a broader spectrum of $n$? This would help readers better understand the effect of $n$.

### Other Comments

- Possible typos or issues:
  - Theorem 3.4: "Gradient … have closed form …" likely should be "has a closed-form …"
  - Line 247–248 (left column): "Figure 3 illustrates empirical trends that are consistent with Theorem 3.3." However, Theorem 3.3 concerns discretization step size, while Figure 3 does not appear to present such information.
  - Table 4: the best HPS results appear to be incorrectly bolded.

- Suggestions for improvement:
  - Figure 4: it would be more informative to annotate the guidance scale $s$ in (a) and $n$ in (b) and (c).
  - The authors may consider reorganizing and refining the experiment section. The current presentation makes it not clear to the readers what research questions this section aims to answer, and which research question(s) each experiment is intended to answer.
  - Appendix A.1, line 620-622: It would be more friendly to readers if the authors explicitly wrote $p(\mathbf x_t|\mathbf x_0) = p(\mathbf x_t|\mathbf x_0, \mathbf c)$, though it was implicitly mentioned via “definition of a forward process”.

**Limitations:**

Limitations like tradeoff between reward and sample diversity should be discussed and explicitly listed.

**Strengths And Weaknesses:**

## Strengths

1. The problem of test-time scaling for diffusion guidance is well-motivated, clearly defined, and important.
2. The proposed method is technically sound and clearly presented, with several supporting theoretical justifications.
3. The method removes the need to compute gradients through both the neural network and the reward function, improving inference efficiency and relaxing the differentiability requirement on the reward.
4. The experimental evaluation is comprehensive, includes reasonable gradient-guidance baselines, and provides multiple ablation studies.

## Weaknesses

1. A direct comparison with SMC methods is not provided (see Q1 below). This weakens the authors’ claim that the proposed method addresses the sensitivity of SMC methods to the number of samples $N$.
2. The choice of reward functions appears to overlap with conditioning objectives (see Q2 below).
3. Qualitative results are limited, as the experiments focus exclusively on text-to-image diffusion models.
4. The paper does not discuss limitations of the proposed approach.
5. Reproducibility: no code implementation or anonymous repository is provided.

---

> ### Author Rebuttal · Authors · 2026-03-31
>
> We sincerely appreciate reviewer **MU56** for the thoughtful evaluation and for recognizing key strengths of our work, including **motivation**, **technical soundness**, and **experimental validation**. We address the comments below.
>
> ---
>
> ### **D1. Direct Comparison to SMC, and BoN**
>
> Compared to vanilla sampling, LiDAR adds cost from generating $n$ lookahead samples, while SMC adds cost to evaluate the decoder and reward model at intermediate time $t$. For fair comparison, we match **memory** and **runtime**, rather than sample counts between $n$ and $N$, since lookahead sampling can be quickly repeated under fixed memory.
>
> |N=4|IR (mean)|Diversity|IR (best)|Time|Memory|
> |-|:-:|:-:|:-:|:-:|:-:|
> |Vanilla|-0.001|0.321|0.749|7.07|8.90|
> |||||||
> |LiDAR (DPM-5 / n=9)|0.211|0.285|0.858|8.27|8.90|
> |SMC ($\lambda$=0.1)|0.062|0.151|0.388|8.97|9.65|
> |SMC ($\lambda$=0.3)|0.138|0.146|0.439|8.97|9.65|
> |SMC ($\lambda$=10)|0.568|0.090|0.692|8.97|9.65|
>
>
> When providing $N=4$ samples to the user, the key metrics are IR (mean) and diversity. Although SMC ($\lambda=10$) achieves high IR (mean), it indeed generates near-duplicate samples (see Figure 2 and Figure 14 for SMC samples), resulting in low diversity and making it unsuitable for providing all $N=4$ samples to the user. The low diversity metric reflects this issue.
>
> Therefore, SMC is better suited for providing a single best sample out of $N=4$ (i.e., combined with BoN), rather than presenting all $N=4$ samples to the user. In this setting, the relevant metric is IR (best), where LiDAR outperforms SMC. As shown in Table 6, comparing SMC (+ BoN) (Line 5) with LiDAR (+ BoN) (Line 4) provides a direct performance comparison across different $N$.
>
> A comparison with Best-K-of-N is shown in the table below. For Vanilla, we generate $N=5$ samples and report the average IR of the top $K$, while LiDAR generates $N=4$ samples at slightly lower cost. From the IR (mean, $K=4$) column, Best-4-of-5 and LiDAR can be directly compared, with LiDAR performing slightly better. Consider providing $K$ samples are presented to the user, LiDAR could be combined with Best-K-of-N and perform better than using Best-K-of-N alone, as shown in IR (mean, K=2), and IR (K=1).
>
>
> |Method|IR (mean, K=4) $(\uparrow)$ |IR (mean, K=2) $(\uparrow)$|IR (K=1) $(\uparrow)$|Time $(\downarrow)$|Memory $(\downarrow)$|
> |-|:-:|:-:|:-:|:-:|:-:|
> |Vanilla|0.209|0.637|0.822|8.83|9.97|
> |LiDAR (DPM-5 / n=9)|**0.211**|**0.670**|**0.876**|**8.27**|**8.90**|
>
> ---
>
> ### **D2. Unconditional Prompts on Diffusion Model**
>
> We test a setup where the diffusion model uses an *unconditional prompt* (for both target and lookahead sampling), while the reward uses a *conditional prompt*.
>
> |Case|Uncond|Cond|
> |-|-|-|
> |Case1|"a photo of one human face"|"a photo of one female human face"|
> |Case2|"a photo of one human face"|"a photo of one elderly human face"|
>
> Applying conditional annotation on the unconditional diffusion model improves performance in both cases, especially in Case 1, where it even outperforms the vanilla conditional diffusion model. In Case 2, gains are smaller, as unconditional sampling produces fewer matching samples (e.g., elderly faces). This suggests LiDAR works better when the supports of unconditional and conditional prompts overlap. **Limited support overlap leads to smaller gains, which we consider a limitation**, though it can be mitigated by increasing $n$.
>
> |Method|Case1|Case2|
> |-|:-:|:-:|
> |Base (Uncond)|-0.62|-1.41|
> |+ LiDAR|**0.86**|**-1.08**|
> ||||
> |Base (Cond)|0.38|1.13|
> |+ LiDAR|**1.41**|**1.36**|
>
> ---
>
> ### **D3. Limited Qualitative Results by Focusing only on T2I Diffusion Model**
>
> In response **C6** to reviewer **ZmXw**, we extend LiDAR to discrete diffusion and conduct molecular generation experiments. We report quantitative results here and will include qualitative analysis (e.g., guided samples, reward statistics) in the final revision.
>
> |$n$ |Num novel $(\uparrow)$|Novel ring count $(\uparrow)$|Total ring count $(\uparrow)$| Time $(\downarrow)$|
> |-|:-:|:-:|:-:|:-:|
> |0|130|2.192|1.753|1.000K|
> |8|162|2.611|2.290|1.002K|
> |16|205|3.034|2.601|1.004K|
> |256|242|3.839|3.272|1.063K|
> |1024|251|3.948|**3.393**|1.250K|
> |4096|**257**|**4.128**|3.067|2.000K|
>
> ---
>
> ### **D4. Reproducibility**
>
> We are organizing the code and obtaining approval for sharing. We will provide anonymized code in the final author response and release the public repository upon acceptance.
>
> ---
>
> ### **D5. Ablation of the Number of Lookahead Samples $n$**
>
> Figure 3 covers a wide range of $n = [0, 3, 9, 16, 25, 50, 200, 400]$ across multiple metrics (ImageReward, CLIP, GenEval) and lookahead solvers (3, 4, 5, 10 steps).
>
> ---
>
> ### **D6. Other Suggestions**
>
> We will address typos and presentation issues in the revision. For Figure 3, different legends correspond to different discretization steps $\delta$, partially aligning with Theorem 3.3. We further clarify Figure 3 with additional analysis on $n$ in response **C1** to reviewer **ZmXw**.

---

> > ### Author Rebuttal · Reviewer_MU56 · 2026-04-01
> >
> > Thanks for the detailed response. The authors have addressed the majority of my concerns. I would highly encourage the authors to add contents D.1 - D.3 in the rebuttal to the paper, with more results and detailed analyses, particularly regarding the arguments around 'diversity'. Looking forward to seeing the actual implementation. I will raise my score accordingly.

---

> > > ### Author Response · Authors · 2026-04-06
> > >
> > > Thank you for your thoughtful feedback and for acknowledging our response. We appreciate your suggestion and will incorporate D.1–D.3, including the additional results and diversity analysis, into the final version.
> > >
> > > ---
> > >
> > > For code sharing, we referred to the rebuttal policy regarding anonymous links and found that links should be used primarily for figures. Therefore, we provide the core parts of the code below in text form, and we will release the full implementation immediately after author notification.
> > >
> > > Our method is implemented by modifying only a single Python file in the official diffusers pipeline. Below, we present a simplified version of the code, including the `get_lookahead_guide` function that computes guidance for latents ($x_t$) at each timestep $t$ using lookahead samples, and how it is integrated into the iterative denoising procedure in `StableDiffusionPipeline`.
> > >
> > > For reference, `self.scale` corresponds to $s$ in Eq. 15, `self.lambda_` corresponds to $\lambda$ in Eq. 16, $n$ denotes the number of lookahead samples, and $N$ denotes the number of target samples being generated. We intentionally keep the implementation minimal for clarity.
> > >
> > > ---
> > >
> > > ```python
> > > def get_lookahead_guide(self, lookaheads, t, latents, lambda_):
> > >     """
> > >     latents: (N=4, 4, 64, 64)
> > >     lookaheads["latents"][0]: (n=50, 4, 64, 64)
> > >     lookaheads["rewards"][0]: (n=50,)
> > >     """
> > >
> > >     device = latents.device
> > >     alpha_prod_t = self.scheduler.alphas_cumprod[t].to(device).float()
> > >
> > >     N = latents.shape[0]
> > >     n = lookaheads["latents"].shape[0]
> > >
> > >     # --------------------------------------------------
> > >     # reshape for broadcasting
> > >     # --------------------------------------------------
> > >     latents = latents.unsqueeze(1)  # (N, 1, 4, 64, 64)
> > >     lookahead_latents = lookaheads["latents"].to(device)  # (n, 4, 64, 64)
> > >     lookahead_latents = lookahead_latents.unsqueeze(0)  # (1, n, 4, 64, 64)
> > >     # --------------------------------------------------
> > >     # potential: (N, n)
> > >     # --------------------------------------------------
> > >     potential = - (latents.float() - (alpha_prod_t ** 0.5) * lookahead_latents) ** 2
> > >     potential = potential / (2 * (1 - alpha_prod_t))
> > >     potential = potential.sum(dim=(2, 3, 4))  # (N, n)
> > >     # --------------------------------------------------
> > >     # weights
> > >     # --------------------------------------------------
> > >     rewards = lookaheads["rewards"].to(device).float()  # (n,)
> > >     rewards = rewards.view(1, n)  # (1, n)
> > >     w_r = F.softmax((lambda_ * rewards + potential), dim=1)  # (N, n)
> > >     w = F.softmax(potential, dim=1)  # (N, n)
> > >
> > >     delta_w = (w_r - w)[..., None, None, None]
> > >     guide = delta_w * lookahead_latents
> > >     guide *= (alpha_prod_t ** 0.5) / (1 - alpha_prod_t)
> > >     guide = guide.sum(dim=1)  # (N, 4, 64, 64)
> > >
> > >     return guide
> > > ```
> > >
> > > ---
> > >
> > > ```python
> > > class StableDiffusionPipeline(
> > >     ...
> > > ):
> > >     ...
> > >
> > >     def __call__(...):
> > >         ...
> > >         ### Sampling xT
> > >         latents = self.prepare_latents(...)
> > >
> > >         ### Iterative Denoising
> > >         with self.progress_bar(total=num_inference_steps) as progress_bar:
> > >             for i, t in enumerate(timesteps):
> > >                 ...
> > >                 ### predict the noise residual
> > >                 noise_pred = self.unet(latents, t,encoder_hidden_states=prompt_embeds,)[0]
> > >                 ### perform cfg
> > >                 ...
> > >                 ### Applying LiDAR
> > >                 ## lookaheads: precomputed samples and rewards from a few-step sampler
> > >                 reward_grad = self.get_lookahead_guide(lookaheads, t, latents, self.lambda_)
> > >                 noise_pred = noise_pred - (1 - self.scheduler.alphas_cumprod[t]) ** (0.5) * reward_grad * self.scale
> > >
> > >                 ### compute the previous noisy sample x_t -> x_t-1
> > >                 step_dict = self.scheduler.step(noise_pred, t, latents, **extra_step_kwargs, return_dict=True)
> > >                 latents = step_dict["prev_sample"]
> > >
> > >         ### Decoding into the image space
> > >         image = self.vae.decode(latents / self.vae.config.scaling_factor)
> > >
> > > ```
> > >
> > > Thank you again for your support.
> > >
> > > Best regards,
> > >
> > > Authors

---

### Official Review · Reviewer_ZmXw · 2026-03-10

**Soundness:** 3
**Presentation:** 3
**Significance:** 3
**Originality:** 3
**Overall Recommendation:** 4
**Confidence:** 4

**Summary:**

The paper introduces Lookahead Sample Reward Guidance (LiDAR), a test-time scaling method designed to align text-to-image diffusion models to the given reward models. Existing gradient guidance methods approximate the Expected Future Reward (EFR) using Taylor expansions, which incur massive computational overhead due to the backpropagation and suffer from approximation errors. LiDAR bypasses this by pre-generating "lookahead" samples using a fast solver (e.g., DPM-solver). By leveraging these lookahead samples, the authors formulate a derivative-free, closed-form Stein score that guides the target sampling process toward high-reward outcomes. Empirically, LiDAR achieves promising results on SD v1.5 and SDXL evaluated on GenEval, ImageReward, and HPS.

**Compliance With Llm Reviewing Policy:**

Affirmed.

**Key Questions For Authors:**

- How robust is LiDAR when the lookahead solver completely fails to generate meaningful structures (e.g., for highly complex compositional prompts)? Is there a mechanism or threshold to reject poor lookahead distributions before they corrupt the target sampling trajectory?
- The lookahead sampler $q(x_0 | c)$ plays a foundational role in the success of LiDAR. I am curious about its behaviour and theoretical limits under two specific settings:
    - What happens if $q(x_0 | c)$ is not a fast solver of the base model, but rather a completely different and more powerful model,  like nano-banana? In this case, how does LiDAR perform
    - $q(x_0 | c)$ is a one-step/few-step distilled model from the base models. Could you expand on how the approximation quality of these distilled models affects the variance of the guidance signal compared to using ODE solvers like DPM? And would the distilled models affect the diversity?
- In Table 6, the authors propose SMC + LiDAR, BoN + LiDAR. Could you explain in detail how these combinations work?
- It would be better to add a discussion in the limitations and potential future work. How LiDAR might be adapted for discrete state spaces where the continuous forward kernel and Stein score formulations do not directly apply.

**Limitations:**

The primary limitations of this work relate to its reliance on fast sampling priors. The approach acts as a "weak-to-strong" generalizer; therefore, it is fundamentally bottlenecked by the capability of the "weak" lookahead sampler to produce at least partially distinguishable reward signals. If the lookahead samples are uniformly poor, the guidance will fail.

**Strengths And Weaknesses:**

**Strongths**:

- The mathematical reformulation of the Expected Future Reward (EFR) to detach dependencies of x_t is interesting.
- Deriving a derivative-free, closed-form guidance mechanism (Theorem 3.4) solves the gradient instability and heavy backpropagation costs of the gradient guidance method.
- LiDAR is very efficient and can maintain good diversity, while it is derivative-free and easily accommodates non-differentiable and black-box reward functions.

**Weakness**:

- The paper lacks a discussion regarding the variance of the Monte Carlo estimator presented in Equation 16. Although this estimator is asymptotically unbiased, Monte Carlo methods in high-dimensional spaces often suffer from high variance. The authors should discuss how this variance impacts the stability of the empirical lookahead reward $\hat{r}_t^\lambda(x_t, c)$ in practice.
- The theoretical analysis in Theorem 3.3 feels somewhat disconnected from the empirical estimator proposed in Equation 16. While the theorem effectively bounds the discretisation error of the few-step lookahead solver, a more impactful theoretical contribution would be to directly analyse the approximation error between the distribution induced by the empirical estimator (Eq. 16) and the true reward-tilted target distribution $p^r_\theta(x_t|c)$.

---

> ### Author Rebuttal · Authors · 2026-03-31
>
> We sincerely appreciate reviewer **ZmXw** for the careful reading of our work and for recognizing its strengths, including the **proposed EFR reformulation**, **derivative-free guidance**, and **efficiency**. We address the comments below.
>
>
> ---
>
> ### **C1. Theoretical Analysis on Number of Lookahead Samples $n$**
>
> In LiDAR, approximation to the target distribution improves as both discretization steps $\delta$ and lookahead samples $n$ increase (Figure 3). Theorem 3.3 shows the gap decreases as $\delta$ increases.
> We further analyze $n$. The empirical lookahead reward $\hat{r}_t^{\lambda}(x_t,c)$ (Eq. 16) converges to the true reward $\tilde{r}_t^{\lambda}(x_t,c)$ (Eq. 12):
>
> **Theorem 3.5.** Under mild conditions, for $\forall t\in[0,1]$ and $c$, the following holds: $\sqrt{n}(\hat{r}_t^{\lambda}(x_t,c)-\tilde{r}_t^{\lambda}(x_t,c))\xrightarrow{d} \mathcal{N}\left(0, \text{Cov}\big(p(x_t|x_0)\exp(\lambda\cdot r(x _0,c))\big) / \big({ \mathbb{E} _{q(x_0|c)} [p(x_t|x_0)]}^2 \exp(2\tilde{r} _t^{\lambda}(x_t,c)) \big) \right)$
>
> *Sketch of proof.* We apply WLLN for the denominator in $\hat{r}_t$ as $n\rightarrow \infty$. Next, we utilize CLT for the remaining term. By putting them together via Slutsky’s theorem and utilizing the delta method.
>
> This implies that larger $n$ yields a smaller variance of reward estimation and improved performance. We will include the theorem above and the proof in the revised version.
>
> ---
>
> ### **C2. Robustness under Poor Lookahead Solver Accuracy**
>
> We compare DPM-3 and DPM-15 lookahead solvers for GenEval categories (The tasks become harder from left to right):
>
> |Lookahead|colors|counting|single_object|two_object|position|color_attr|All|
> |-|:-:|:-:|:-:|:-:|:-:|:-:|:-:|
> |Vanilla|0.524|0.443|0.283|-0.155|-0.352|-0.562|0.001|
> |DPM-3|0.777| 0.542|0.458|0.153|-0.067|-0.023|0.288|
> |DPM-15|0.832|0.574|0.518|0.321|0.048|0.170|0.397|
> |||||||||
> |Gap|0.054|0.032|0.060|0.168|0.116|0.194|-|
>
> - The performance gap increases with complexity, highlighting the value of stronger lookahead solvers. This can be mitigated by adaptively adjusting accuracy, making complexity-aware strategies promising. (Note that the lookahead accuracy is fully controllable)
>
> - Even with DPM-3, we observe consistent gains across all categories, indicating robustness to imperfect lookahead. In Eq. 17, when $\lambda$ is large, $w^r$ becomes one-hot, selecting the highest-reward sample. If $x_t$ is already closest to it, the guidance $w^r - w$ vanishes. Thus, Eq. 17 primarily drives $x_t$ to be closest to the highest-reward lookahead sample among all lookahead candidates, and is less likely to corrupt it than the vanilla approach.
>
> ---
>
> ### **C3. Using Powerful Lookahead Solver**
>
> Due to resource constraints, we plan to use a FLUX-based model for lookahead sampling with SD v1.5 and will report results in the author’s final response. (If the reviewer responds, the system allows us to provide a response.)
>
> ---
>
> ### **C4. Analysis on Distillation Lookahead Solver**
>
> As discussed in **A3** to reviewer **fF52**, theory suggests that as the distillation model better approximates $p_{\theta}$, it more accurately models the target distribution. We compare DPM-8, LCM-4, and DMD-1 lookahead samplers (n=100) under SDXL. Higher lookahead diversity leads to higher target diversity. Distillation models such as DMD-1 show reduced diversity due to adversarial training, which can also reduce target sample diversity.
>
> |Method|Lookahead Diversity $(\uparrow)$|Target Diversity $(\uparrow)$|Target IR $(\uparrow)$|
> |-|:-:|:-:|:-:|
> |LiDAR (DPM-8)|0.363|0.250|1.014|
> |LiDAR (LCM-4)|0.342|0.238|1.007|
> |LiDAR (DMD-1)|0.236|0.233|1.006|
>
> ---
>
> ### **C5. Experimental setting for SMC + LiDAR, BoN + LiDAR**
>
> For SMC (Lines 168–185), LiDAR-guided samples are used when drawing proposal particles. For BoN, we select the highest-reward sample among $N$ LiDAR-generated samples. Although lookahead adds overhead, Figure 7 shows that LiDAR achieves a better Pareto trade-off between reward and wall-clock time.
>
> ---
>
> ### **C6. Extension to Discrete Diffusion**
>
> The Stein score in Eq. 5 can be rewritten in a denoising form and extended to discrete domains:
> $$
> p_{\theta}^r(x_{t-1}|c, x_t) \propto p_\theta(x_{t-1}|c, x_t)\cdot \exp\big(r_{t-1}^{\lambda}(x_{t-1},c) - r_t^{\lambda}(x_t,c)\big)
> $$
> We use a QM9-trained molecular discrete diffusion model with ring count (structural complexity)  as the reward and generate $N=1024$ samples. We report novelty, ring count, average ring count, and runtime relative to vanilla. As the number of lookahead samples $n$ increases, all metrics improve gradually. The lookahead sampler is 4× faster than the target sampler.
>
> |$n$ |Num novel $(\uparrow)$|Novel ring count $(\uparrow)$|Total ring count $(\uparrow)$| Time $(\downarrow)$|
> |-|:-:|:-:|:-:|:-:|
> |0|130|2.192|1.753|1.000K|
> |8|162|2.611|2.290|1.002K|
> |16|205|3.034|2.601|1.004K|
> |256|242|3.839|3.272|1.063K|
> |1024|251|3.948|**3.393**|1.250K|
> |4096|**257**|**4.128**|3.067|2.000K|

---

> > ### Author Rebuttal · Reviewer_ZmXw · 2026-04-04
> >
> > Thanks for the reply. The result looks promising. I intend to accept the paper

---

> > > ### Author Response · Authors · 2026-04-06
> > >
> > > Thank you for your positive feedback. We are glad that you found the results promising, and we truly appreciate your support. We provide our response to the previously unanswered question below.
> > >
> > > ---
> > >
> > >
> > > ### **C3. Using powerful lookahead solver**
> > >
> > > *Q. What happens if $q(x_0|c)$ is not a fast solver of the base model, but rather a completely different and more powerful model?*
> > >
> > > We conduct experiments using SDv1.5 (0.9B) as the base model $p_{\theta}(x_0|c)$ and a much stronger lookahead sampler $q(x_0|c)$, Hyper-FLUX.1-dev-8step (12B) [1]. The table below compares the results with those obtained when using SDv1.5 with a DPM-5step sampler as the lookahead sampler, across different values of $n$.
> > >
> > > | Method | n | | DPM-5 | || | Hyper-FLUX-8 | |
> > > |-|:-:|:-:|:-:|:-:|:-:|:-:|:-:|:-:|
> > > | | | IR ($\uparrow$) | CLIP ($\uparrow$) | HPS ($\uparrow$) || IR ($\uparrow$) | CLIP ($\uparrow$) | HPS ($\uparrow$) |
> > > | SDv1.5| 0 | -0.001 | 0.271 | 0.263 || -0.001 | 0.271 | 0.263 |
> > > | LiDAR | 3 | **0.172** | **0.275** | **0.274** || 0.051 | 0.273 | 0.263 |
> > > | LiDAR | 9 | **0.211** | **0.276** | **0.274** || 0.122 | 0.273 | 0.263 |
> > > | LiDAR | 25 | **0.242** | **0.276** | **0.275** || 0.188 | 0.274 | 0.265 |
> > > | LiDAR | 50 | **0.384** | **0.278** | **0.276** || 0.196 | 0.275 | 0.266 |
> > >
> > > As $n$ increases, all metrics improve in both cases. However, using the base model with a few-step lookahead sampler shows a steeper performance gain compared to using a completely different lookahead sampler. The theoretical analysis below may explain this phenomenon.
> > >
> > > In the proof of Theorem 3.3, by combining Eq. 21, Eq. 28, and Eq. 35, we obtain the following bound (assuming no prior gap for simplicity). We can express the gap between the target distribution $p_{\theta}^{r}(x_0|c)$ and the lookahead induced distribution $\tilde{p_{\theta}}^{r}(x_0|c)$ in terms of the gap between the base model distribution $p_{\theta}(x_0|c)$ and the lookahead distribution $q(x_0|c)$:
> > >
> > > $$
> > > D _{TV}\big(p _{\theta}^{r}(x_0|c) | \tilde{p} _{\theta}^{r}(x_0|c)\big)
> > > \le
> > > \frac{2M}{m}D _{TV}\big(p _{\theta}(x_0|c)|q(x_0|c)\big)
> > > +
> > >  \sqrt{\frac{M}{2m} D _{KL}\big(p _{\theta}(x_0|c)|q(x_0|c)\big)}
> > > $$
> > >
> > > We conjecture that the lookahead distribution obtained from a few-step version of the base model is closer to the base model than that from a completely different model. As a result, it produces samples that are closer to the target distribution, which may explain the observed results.
> > >
> > > Intuitively, in target sampling (phase 2), the base model receives a signal to move closer to high-reward lookahead samples and farther from low-reward ones. If the current denoising sample $x_t$ becomes close to images that are very different from those of the base model, it may move into regions where the base model cannot properly denoise, which could negatively affect sampling.
> > >
> > > When we already have a base model, it is more natural to obtain a weaker version of it (e.g., a few-step solver or a distilled model) than to obtain a much stronger lookahead model. Therefore, the weak-to-strong interpretation proposed in our work is more practical for real use cases, and it also leads to better performance.
> > >
> > > [1] (NeurIPS 2024) Hyper-SD: Trajectory Segmented Consistency Model for Efficient Image Synthesis
> > >
> > > ---
> > >
> > > Thank you again for your support.
> > >
> > > Best regards,
> > >
> > > Authors

---

### Official Review · Reviewer_p1ba · 2026-03-12

**Soundness:** 2
**Presentation:** 2
**Significance:** 3
**Originality:** 3
**Overall Recommendation:** 4
**Confidence:** 4

**Summary:**

The study explores a critical question in generative modeling: how to effectively scale test-time compute to guide diffusion models toward higher human-aligned rewards without the computational bottleneck of sequential neural backpropagation. The authors claim to achieve this by approximating the Expected Future Reward (EFR) at intermediate timesteps using marginal samples generated from a pre-trained diffusion model. By using forward samples to estimate the expected reward and guiding the intermediate noisy states accordingly, the paper proposes a test-time scaling method intended to improve alignment in text-to-image (T2I) models.

**Compliance With Llm Reviewing Policy:**

Affirmed.

**Final Justification:**

The additional comparisons and detailed explanation clarified the distinction from recent methods and helped me better understand the proposed approach. Thus, I raise my score to 4 accordingly.

**Key Questions For Authors:**

- Could the authors explain the mathematical and methodological difference to previous works including [General-Test-time] and [FKC]?

- The authors claim to introduce a novel approach for estimating the expected future reward, yet the recent literature already contains several advanced, established approximations for the value function gradient $\nabla V_t(x)$ (e.g., [GLASSFlow], [MFM], and [AM]). How does your proposed method theoretically compare to these established approximation schemes, and why were direct empirical comparisons against these highly relevant baselines (particularly [FKC]) omitted from the experiments?

- Could you provide a detailed analysis of the wall-clock inference time and performance trade-off of your method compared to previous test-time scaling approaches?

**Limitations:**

Refer to the weakness section.

**Strengths And Weaknesses:**

**Strength**
- The motivation is sound. Improving test-time compute scaling for diffusion models to maximize human-aligned rewards is a highly relevant and active area of research, especially for large-scale T2I models where backpropagating through the network is expensive.

**Weakness**

The core methodology of this paper is fundamentally a simple application of the Feynman-Kac formula to diffusion models, which has already been extensively explored. Fine-tuning a diffusion model can be formally analyzed by solving a Linear-Quadratic (LQ) stochastic control problem given the memoryless dynamics (of the pretrained model) [AM]. In this framework, the optimal guidance is dictated by the gradient of the value function, $\nabla V_t(x)$. Estimating this expected future reward via forward sample rollouts is exactly what the Feynman-Kac formula dictates. The paper presents this as a novel mechanism, but it lacks any new mathematical or methodological innovation over standard stochastic control principles.

The paper completely misses the rich, established literature that already applies these exact Feynman-Kac principles to generative models. Papers such as [General-Test-time], [ELEGANT], [FKC] (and its variants), [GLASSFlow], and [MFM] have heavily explored this space. Furthermore, simply applying this formulation to test-time scaling is explicitly discussed and established in the [FKC] and [General-Test-time] literature, making the proposed approach virtually identical to existing methods.

The task of approximating the gradient of the value function $\nabla V_t(x)$ has seen significant recent innovation. Methods like [General-Test-time], [Tilt], and [MFM] have already established rigorous approximations for this exact guidance problem. The proposed method does not offer a novel theoretical position or a superior approximation scheme compared to these existing works.

Given the lack of theoretical novelty, the paper must rely on exceptional empirical results to justify acceptance; however, the reported performance falls significantly short. The proposed method shares profound similarities with existing approaches, featuring only minor methodological differences. Consequently, it is imperative to demonstrate a clear empirical advantage over these prior works, yet I do not see any such advantage compared to established test-time scaling and Feynman-Kac approaches (e.g., [General-Test-time], [FKC]). Crucially, the experiments omit comparisons against the most relevant mathematical baselines.

**References**

[AM] Adjoint Matching, ICLR, 2025.

[General-Test-time] A General Framework for Inference-time Scaling and Steering of Diffusion Models, ICML, 2025.

[ELEGANT] Fine-Tuning of Continuous-Time Diffusion Models as Entropy-Regularized Control, 2024.

[FKC] Feynman-Kac Correctors in Diffusion: Annealing, Guidance, and Product of Experts, ICML, 2025.

[GLASSFlow] GLASS Flows: Efficient Inference for Reward Alignment of Flow and Diffusion Models, ICLR, 2026.

[MFM] Meta Flow Maps enable scalable reward alignment, 2026.

[Tilt] Tilt Matching for Scalable Sampling and Fine-Tuning, 2025.

(Note to Authors/Area Chair: The citations provided above are strictly for reference to highlight the established, foundational stochastic control literature in this domain. My evaluation strictly adheres to the conference review policy and does not penalize the submission based on unreferred preprints or concurrent work published within the last 3 months.)

---

> ### Author Rebuttal · Authors · 2026-03-31
>
> We thank reviewer **p1ba** for highlighting important related work. We clarify the novelty of our work.
>
> ---
>
> ### **B. Taxonomy for Reward-tilted Sampling**
>
> Our manuscript describes key components (*B-1, B-2, B-3*) for reward-tilted sampling. The components are orthogonal, and prior works primarily focus on one part.
> ### *B-1. Formulation of Expected Future Reward (EFR) (Eq.5)*:
> - Consideration: What form of EFR satisfies the target distribution $p_{\theta}^r(x_0|c)$ at $t=0$ (i.e., the marginal at time $t$)?
> - B-1.1. [Elegant]: Proposes $p_{\theta}^{r}(x_t|c) \propto p_{\theta}(x_t|c) \mathbb{E}_{p(x_0|x_t)} [\exp(\lambda r(x_0))]$
>     - [AM]: Provides stochastic control interpretation of it.
>     - **LiDAR adopts it**
> - B-1.2. [FKC]: Proposes $p_{\theta}^{r}(x_t|c)\propto p_{\theta}(x_t|c) \exp((\lambda r(x_t))$. It requires a reward function that evaluates $x_t$ at all $t$; however, no concrete instantiation is provided.
> - B-1.3. [General-Test-time]: Generalizes discrete-time EFR formulations with instantiations. It defines $p^r_{\theta}(x_T, …, x_t |c) \propto p_{\theta}(x_T, …, x_t |c) \prod_{s=T}^{t}\ G_t(x_T, …, x_s,c)$, and proposes $G_t$ such that $\prod_{t=T}^{0}\ G_t (x_T, …, x_s, c)=\exp(\lambda r(x_0, c))$.
> ### ***B-2. Computation of EFR (Section 2.2.2): The area where LiDAR claims novelty.***
> - B-2.1. Taylor approximation with Tweedie’s formula (Eq.7)
>    - As it avoids backward rollout, it is efficient and used as a baseline.
> - B-2.2. Backward rollout: (Relation $x_0↔x_t$ is given by $p_{\theta}(x_0|x_t)$)
>     - B-2.2.0. Eq.6: At each $t$, EFR is computed by generating $n$ samples of $x_0$ from $x_t$ via iterative denoising.
> $$
> r_t^{\lambda}(x_t, c) \approx \log  \frac{1}{n} \sum_{i=1}^{n} \exp\big( \lambda r(x_0^{i}, c) \big) , \quad \textcolor{red}{x_0^i \sim p^{SDE}_\theta(x_0 \mid x_t, c)}
> $$
>     - B-2.2.1. [GLASSFlow]: Replaces sampling to $\textcolor{red}{ x_0^i\sim p ^{ODE}_\theta (x_0|x_t,c)}$ to reduce steps.
>     - B-2.2.2. [MFM]: Replaces sampling to one-step model $\textcolor{red}{ x_0^i\sim p_{\phi}^{One-step}(x_0|x_t,c)}$.
> - B-2.3. **Forward rollout (Ours)**: (Relation $x_0↔x_t$ is given by $p(x_t|x_0)$, that does not require neural denoising.)
>     -  (Theorem 3.1):  LiDAR computes EFR using marginal samples $q(x_0|c)$, avoiding per-timestep sampling:
> $$
> \log\frac{1}{n} \sum _{i=1}^{n} \left[
> \frac{\textcolor{red}{p(x _t |\hat{x} _0^i)}}
> {\frac{1}{n} \sum _{j=1}^{n}\textcolor{red}{p(x_t | \hat{x} _0^j)}}
> \exp\left(\lambda r(\hat{x} _0^i, c)\right)
> \right], \quad \textcolor{blue}{\hat{x}_0^{i} \sim q(x_0|c)}
> $$
>     - (Definition 3.2, Theorem 3.3): LiDAR proposes a fast ODE or a distillation sampler to make the forward rollout even faster.
>
> ### *B-3. Sampling with computed EFR (Section 2.2.3)*
> - B-3.1. Guidance
>    - [UG, DATE]: Guidance for Taylor in B-2.1.
>    - [Tilt]: Backward-free guidance for backward rollout in B-2.2.
>    - (Theorem 3.4): Backward-free guidance for forward rollout in B-2.3.
> - B-3.2. SMC (Lines 168-186)
>    - Applicable regardless of *B-2*, but suffers from particle collapse.
> ---
> ### **BQ.0. Benefit of LiDAR among *B.2***
>
> LiDAR shows the lowest added neural cost, with best qualitative properties within the blocks; see Fig. 8 for Taylor error ($N$: # of target samples).
>
> |Method| $s_{\theta}$|$r$|Training|Neural Backward|Taylor Error|
> |-|:-:|:-:|:-:|:-:|:-:|
> |Eq.6|$\frac{NnT(T+1)}{2}$|$NnT$|X|O|X|
> |Eq.6 + [Tilt]|$\frac{NnT(T+1)}{2}$|$NnT$|X|X|X|
> |LiDAR ($q = p_{\theta}$)|$nT$|$n$|X|X|X|
> |||||
> |[GLASSFlow]|$\frac{Nn\delta(T+1)}{2}$|$NnT$|X|O|X|
> |LiDAR ($q = p^{ODE}_{\theta}$)|$n\delta$|$n$|X|X|X|
> |||||
> |[MFM]|$NnT$|$NnT$|O|O|X|
> |LiDAR ($q = p^{One-step}_{\phi}$)|$n$|$n$|O|X|X|
> |||||
> |Taylor|x|$NT$|X|O|O|
>
> ---
>
> ### **BQ.1 Positioning of LiDAR**
> - While [General-Test-time, FKC] focus on *(B-1: Formulation of EFR)*, LiDAR focuses on *(B-2: Computation of EFR)*.
> - Concurrent preprints [GLASSFlow, MFM, Tilt] improve (B-2.2: backward rollout), while **our key contribution is introducing *(B-2.3: forward rollout)***, which changes how the relation $x_0 ↔ x_t$ is computed. **None of the referred works appears to provide our forward rollout formulation.**
> ---
> ### **BQ.2 Empirical Comparison**
> - Section 3.5 of [FKC] does not specify a concrete form of the reward at $t$, making comparison difficult. [FKC] does not perform experiments where the reward is only available at $t=0$, as in our setup (line 127).
> - [General-Test-time] is indeed included as a baseline. In Figure 7 and Table 6, “SMC” corresponds to [General-Test-time]. The leftmost plot in Figure 7 shows results over $N$ samples. LiDAR performs better under equal wall-clock time. More results are provided in **D1 of reviewer MU56**.
> ---
> ### **BQ.3 Performance Efficiency Tradeoff**
> - Figure 4 (b, c) compares under guidance sampling, LiDAR improves performance with more lookahead samples at a higher cost. Figure 7 compares under SMC. Table 7 breaks down time for lookahead, reward annotation, and target sampling.

---

> > ### Author Rebuttal · Reviewer_p1ba · 2026-04-03
> >
> > The additional comparisons and detailed explanation clarified the distinction from recent methods and helped me better understand the proposed approach. Thank you for the clarification. I will raise my score to 4 accordingly.

---

> > > ### Author Response · Authors · 2026-04-06
> > >
> > > Thank you for your positive feedback and for acknowledging that your concerns have been adequately addressed. We are glad that our additional comparisons and explanations helped clarify our contributions. We will also include this content in the main text.
> > >
> > > Thank you again for your support.
> > >
> > > Best regards,
> > >
> > > Authors

---

### Official Review · Reviewer_fF52 · 2026-03-13

**Soundness:** 3
**Presentation:** 3
**Significance:** 3
**Originality:** 3
**Overall Recommendation:** 5
**Confidence:** 3

**Summary:**

The proposed LiDAR(Lookahead Sample Reward Guidance) framework systematically addresses the critical bottlenecks in prior methods. It eliminates the neural dependency and expensive backpropagation required by gradient guidance methods, bypasses the restrictive black-box reward model constraints and severe particle collapse issues of Sequential Monte Carlo (SMC), and prevents the approximation error from exploding under strong guidance scales of Taylor Approximation.  The paper provides solid mathematical justifications for its design choices, including a closed-form, derivative-free expression for the empirical lookahead reward gradient. Furthermore, it establishes a formal error bound ($O(1/\sqrt{\delta})$), proving that the discrepancy between the target reward-tilted distribution and the lookahead sampler decreases predictably as the number of discretization steps ($\delta$) increases. By utilizing fast solvers or distilled models (e.g., DPM, LCM, DMD) for lookahead sampling, LiDAR matches the top performance of state-of-the-art gradient guidance methods. However, the evaluated backbones (SD 1.5, SDXL) are somewhat outdated, and additional expirements are required.

**Compliance With Llm Reviewing Policy:**

Affirmed.

**Final Justification:**

All my concerns have been adequately addressed, and I recommend acceptance on condition that they add A1~A5 to the camera ready version.

**Key Questions For Authors:**

1. Could the authors provide experimental results using a significantly smaller number of pre-samples (smaller $n$)? This would better demonstrate the robustness and the practical lower-bound efficiency of the proposed LiDAR method.
2. Given the empirical success of standard distilled models in your framework, exploring advanced models that combine distillation with preference/RL tuning (e.g., Flash-DMD[1], Hyper-SD[2]) as the lookahead solver would be highly exciting. Have the authors considered or tested these variants?

[1] Flash-DMD: Towards High-Fidelity Few-Step Image Generation with Efficient Distillation and Joint Reinforcement Learning

[2] Hyper-SD: Trajectory Segmented Consistency Model for Efficient Image Synthesis

**Limitations:**

Please refer to the weakness.

**Strengths And Weaknesses:**

### Strengths
1. The proposed LiDAR(Lookahead Sample Reward Guidance) framework systematically addresses the critical bottlenecks in prior methods. It eliminates the neural dependency and expensive backpropagation required by gradient guidance methods, bypasses the restrictive black-box reward model constraints and severe particle collapse issues of Sequential Monte Carlo (SMC), and prevents the approximation error from exploding under strong guidance scales of Taylor Approximation.
2. The paper provides solid mathematical justifications for its design choices, including a closed-form, derivative-free expression for the empirical lookahead reward gradient. Furthermore, it establishes a formal error bound ($O(1/\sqrt{\delta})$), proving that the discrepancy between the target reward-tilted distribution and the lookahead sampler decreases predictably as the number of discretization steps ($\delta$) increases.
3. By utilizing fast solvers or distilled models (e.g., DPM, LCM, DMD) for lookahead sampling, LiDAR matches the top performance of state-of-the-art gradient guidance methods on S to levels comparable with vanilla sampling.

### Weakness
1. The evaluated backbones (SD 1.5, SDXL) are somewhat outdated, lacking experiments on recent Flow Matching-based models (e.g., SD3, FLUX). Furthermore, the evaluation metrics are relatively limited; incorporating broader preference metrics like Pick-a-Pic and MPS [1] would significantly strengthen the empirical claims.
2. Utilizing distilled models (e.g., DMD) for lookahead sampling lacks theoretical justification. Since DMD relies on adversarial training with real data, its output distribution deviates significantly from the original SDXL. Please provide theoretical support or discussion on how this distribution mismatch impacts the Expected Future Reward (EFR) approximation.
3.  Could the authors provide experimental results using a significantly smaller number of pre-samples (smaller $n$)? This would better demonstrate the robustness and the practical lower-bound efficiency of the proposed LiDAR method.
4. Given the empirical success of standard distilled models in your framework, exploring advanced models that combine distillation with preference/RL tuning (e.g., Flash-DMD[2], Hyper-SD[3]) as the lookahead solver would be highly exciting. Have the authors considered or tested these variants?

[1] Learning multidimensional human preference for text-to-image generation

[2] Flash-DMD: Towards High-Fidelity Few-Step Image Generation with Efficient Distillation and Joint Reinforcement Learning

[3] Hyper-SD: Trajectory Segmented Consistency Model for Efficient Image Synthesis

---

> ### Author Rebuttal · Authors · 2026-03-31
>
> We sincerely appreciate reviewer **fF52** for the deep understanding of our work and for recognizing its strengths, including the **strong theoretical justifications**, and **competitive performance**. We address the reviewer’s comments below.
>
> ---
>
> ### **A1. Pick-a-Pic and MPS**
>
> Thank you for the helpful suggestion. For the results in Table 2, we additionally evaluate Pick-a-Pic Score and MPS for both an efficient backbone (SD v1.5 with DDIM 50 steps) and a powerful backbone (SDXL with DDPM 100 steps), comparing the baselines and LiDAR as shown below. LiDAR achieves the highest scores among all methods on both metrics. We conjecture that lookahead samples with higher ImageReward annotations also tend to achieve higher Pick-a-Pic and MPS scores (similar to Figure 6), which likely explains these results. These metrics will be included in the final version.
> | Method | Pick-a-Pic ($\uparrow$) | MPS ($\uparrow$) |
> |-|:-:|:-:|
> | SDv1.5 (DDIM) | 21.68 | 10.49 |
> | SDv1.5 (200-step) | 21.71 | 10.52 |
> | UG | 20.84 | 8.94 |
> | DATE | 21.73 | 10.57 |
> | LiDAR (DPM-5 / n=50) | **21.89** | **10.94** |
> ||||
> | SDXL (DDPM) | 22.88 | 12.23 |
> | SDXL (250-step) | 22.93 | 12.29 |
> | UG | 22.59 | 11.86 |
> | DATE | 22.88 | 12.24 |
> | LiDAR (DMD1 / n=100) | **22.97** | **12.50** |
>
> ---
>
> ### **A2. FLUX Backbone**
>
> We verify LiDAR on the recent FLUX.1 [dev] (12B). Using Hyper-FLUX-8step as the lookahead sampler, performance consistently improves as the number of lookahead samples increases.
> |Num_lookaheads|IR ($\uparrow$)|CLIP ($\uparrow$)|GenEval ($\uparrow$)|
> |-|:-:|:-:|:-:|
> |Vanilla (n=0)|1.019|0.282|0.645|
> |n=3|1.038|0.282|0.663|
> |n=50|1.114|0.283|**0.668**|
> |n=100|**1.198**|**0.284**|0.667|
>
> ---
>
> ### **A3. Theoretical Analysis on Lookahead Sampling with Distillation Models**
>
> Theorem 3.3 provides a concrete instantiation by specifying $q(x_0|c)$ as a few-step discretized distribution. The more general case is already implicitly covered in its proof. By combining Eq. 21, Eq. 28, and Eq. 35, we obtain the following bound (Assume no prior gap for simplicity):
> $$
> D _{TV}\big(p _{\theta}^{r}(x_0|c) | \tilde{p} _{\theta}^{r}(x_0|c)\big)
> \le
> \frac{2M}{m}D _{TV}\big(p _{\theta}(x_0|c)|q(x_0|c)\big)
> +
>  \sqrt{\frac{M}{2m} D _{KL}\big(p _{\theta}(x_0|c)|q(x_0|c)\big)}
> $$
>
> In other words, the gap between the target distribution $p^r_{\theta}(x_0|c)$ and the lookahead-induced distribution $\tilde{p}^{r}(x_0|c)$ can be decomposed in terms of the discrepancy between the pretrained diffusion model $p_{\theta}$ and the lookahead distribution $q$. When $q$ is instantiated as a distilled model (e.g., LCM), it is explicitly trained to approximate $p_{\theta}$, making the distillation objective aligned with minimizing our error gap. As pointed out, in cases such as DMD where $q$ is trained to match a mixture of $p_{\theta}$ and $p_{data}$, we note that $p_{\theta}$ itself is trained to approximate $p_{data}$. Under the ideal assumption that $p_{\theta} \approx p_{data}$, $q$ can still be viewed as effectively approximating $p_{\theta}$, and thus the resulting error bound remains largely helpful with the objective of distillation.
>
> ---
>
> ### **A4. Performance of Small Lookahead Samples (n)**
>
> We already analyze this in detail (Fig. 3, Table 3), showing strong gains even at small $n$. According to Eq. 17, when $n=1$, $w^r$ and $w$ are always the same, so no guidance signal is produced. However, when $n\geq2$, a meaningful guidance signal is generated. In Figure 3, we report results on three metrics across a wide range of $n$=[0 (identical to vanilla), 3, 9, 16, 25, 50, 200, 400] using SD v1.5. Table 3 further presents the exact values, along with results on a DPO-tuned model. As $n$ increases from small values, we observe a steep performance gain, followed by gradual saturation. We further provide the scaling behavior under small $n$ of IR metric, including SDXL and FLUX, as follows:
>
> | Model | n=0 | 3 | 9 | 50 |
> |-|:-:|:-:|:-:|:-:|
> |SDv1.5|-0.001 | 0.172 |0.211|0.384|
> |SDv1.5 (DPO-tuned)|0.106|0.268|0.296|0.445|
> |SDXL|0.722|0.825|0.892|0.994|
> |FLUX|1.019|1.038|1.045|1.114|
>
> ---
>
> ### **A5. Lookahead Sampling with RL-Tuned Distillation Models**
>
> Among the suggested methods (Flash-DMD and Hyper-SD), we use Hyper-SD as it is open-source. Using a 1-step RL-tuned model with ImageReward as the lookahead solver, it achieves worse ImageReward than DMD, though it still improves over the vanilla baseline as $n$ increases. As discussed in **A3**, performance depends on how well $q$ approximates $p_{\theta}$, and RL-tuned samplers $q$ may deviate from $p_{\theta}$, leading to degradation. As illustrated in Figure 1(b), while LiDAR guides samples away from low-quality regions toward high-quality ones, overly reward-biased lookahead samples may reduce the ability to escape poor regions.
>
> | Backbone (Lookahead)|n = 0 (vanilla)|3|16|100|
> |-|:-:|:-:|:-:|:-:|
> | SDXL (w/ DMD-1)|0.722|**0.835**|**0.930**|**1.006**|
> | SDXL (w/ Hyper-SD-1)|0.722|0.818| 0.882|0.993|

---

> > ### Author Rebuttal · Reviewer_fF52 · 2026-04-05
> >
> > All my concerns have been adequately addressed, and I recommend acceptance on condition that they add A1~A5 to the camera-ready version.

---

> > > ### Author Response · Authors · 2026-04-06
> > >
> > > Thank you for your positive feedback again and for acknowledging that your concerns have been addressed. We sincerely appreciate your careful review and constructive suggestions throughout the process. We will make sure to incorporate A1–A5 into the camera-ready version as recommended.
> > >
> > > Thank you again for your support.
> > >
> > > Best regards,
> > >
> > > Authors

---

### Decision · Program_Chairs · 2026-04-30

**Decision:**

Accept (spotlight)

**Comment:**

The reviewers have reached a clear consensus to accept this paper. They highlighted the novel closed-form EFR formulation that eliminates expensive neural backpropagation, which is a key limitation of prior gradient guidance methods. The proposed LiDAR sampling with lookahead reward guidance can achieve substantial performance gains using only three samples and a 3-step lookahead. Reviewers agree the strong empirical results across multiple tasks and the elegant test-time scaling behavior. While some suggested additional ablation studies, these do not hurt the core contributions of this work.

Overall, the work made a significant contribution to reward-guided diffusion sampling. I recommend Accept.